Clinical Science and Epidemiology

# Emergence of *mcr-9.1* in Extended-Spectrum-*β*-Lactamase-Producing Clinical *Enterobacteriaceae* in Pretoria, South Africa: Global Evolutionary Phylogenomics, Resistome, and Mobilome

John Osei Sekyere,[a] Nontuthuko E. Maningi,[a,b] Lesedi Modipane,[a] Nontombi Marylucy Mbelle[a,c]

[a]Department of Medical Microbiology, Faculty of Health Sciences, University of Pretoria, Pretoria, South Africa
[b]Department of Microbiology, School of Life Sciences, College of Agriculture, Engineering and Science, University of KwaZulu-Natal, Westville, South Africa
[c]National Health Laboratory Services, Tshwane Academic Division, Department of Medical Microbiology, University of Pretoria, Pretoria, South Africa

**ABSTRACT** Extended-spectrum-*β*-lactamase (ESBL)-producing *Enterobacteriaceae* are critical-priority pathogens that cause substantial fatalities. With the emergence of mobile *mcr* genes mediating resistance to colistin in *Enterobacteriaceae*, clinicians are now left with few therapeutic options. Eleven clinical *Enterobacteriaceae* strains with resistance to cephems and/or colistin were genomically analyzed to determine their resistomes, mobilomes, and evolutionary relationships to global strains. The global phylogenomics of *mcr* genes and *mcr-9.1*-bearing genomes were further analyzed. Ten isolates were ESBL positive. The isolates were multidrug resistant and phylogenetically related to global clones but distant from local strains. Multiple resistance genes, including $bla_{CTX-M-15}$ $bla_{TEM-1}$, and *mcr-9.1*, were found in single isolates; IS*Ec9,* IS*19,* and Tn*3* transposons bracketed $bla_{CTX-M-15}$ and $bla_{TEM-1}$. Common plasmid types included IncF, IncH, and ColRNAI. *mcr-9* was of close sequence identity to *mcr-3, mcr-5, mcr-7, mcr-8,* and *mcr-10*. Genomes bearing *mcr-9.1* clustered into six main phyletic groups (A to F), with those of this study belonging to clade B. *Enterobacter* species and *Salmonella* species are the main hosts of *mcr-9.1* globally, although diverse promiscuous plasmids disseminate *mcr-9.1* across different bacterial species. Emergence of *mcr-9.1* in ESBL-producing *Enterobacteriaceae* in South Africa is worrying, due to the restricted therapeutic options. Intensive One Health molecular surveillance might discover other *mcr* alleles and inform infection management and antibiotic choices.

**IMPORTANCE** Colistin is currently the last-resort antibiotic for difficult-to-treat bacterial infections. However, colistin resistance genes that can move from bacteria to bacteria have emerged, threatening the safe treatment of many bacterial infections. One of these genes, *mcr-9.1*, has emerged in South Africa in bacteria that are multidrug resistant, further limiting treatment options for clinicians. In this work, we show that this new gene is disseminating worldwide through *Enterobacter* and *Salmonella* species through multiple plasmids. This worrying observation requires urgent action to prevent further escalation of this gene in South Africa and Africa.

**KEYWORDS** ESBLs, *Enterobacteriaceae*, *mcr-9.1*, South Africa, critical pathogens, colistin resistance

Address correspondence to John Osei Sekyere, jod14139@gmail.com.

mcr-9.1 has emerged in ESBL-positive Enterobacteriaceae in South Africa, restricting therapeutic options and putting hospitalized patients at risk of untreatable infections.

*E*nterobacteriaceae producing extended-spectrum *β*-lactamases (ESBLs) are categorized as critical priority 1 pathogens requiring urgent attention with regard to the development of new antimicrobials (1). Owing to the fatalities caused by ESBL-positive *Enterobacteriaceae*, carbapenems were introduced into clinical practice as last-resort antibiotics, which unfortunately led to the proliferation of strains expressing carbap-

TABLE 1 Demographic and genomic characteristics of the isolates

| Sample code | Accession no. | Age (yr) | Gender | Referring hospital | ESBL | Species | No. of contigs | Size (Mb) | % G+C | No. of RNAs | No. of tRNAs | No. of coding sequences | $N_{50}$ | $L_{50}$ | Coverage (×) | No. of CRISPR arrays |
|---|---|---|---|---|---|---|---|---|---|---|---|---|---|---|---|---|
| CF003 | NXKE00000000 | 25 | F | Tshwane Academic | + | C. freundii | 235 | 5.1 | 51.8 | 120 | 79 | 4,873 | 250,864 | 7 | 99 | 0 |
| CF004 | NXKF00000000 | 34 | M | Tshwane Academic | − | C. freundii | 100 | 4.9 | 51.7 | 85 | 64 | 4,576 | 250,864 | 7 | 102 | 0 |
| EC001 | NXIK00000000 | 42 | M | Tshwane Academic | + | E. hormaechei | 169 | 5.0 | 54.9 | 101 | 71 | 4,762 | 190,180 | 8 | 94 | 0 |
| EC009 | NXIL00000000 | 51 | F | Tshwane Academic | + | E. hormaechei | 446 | 4.9 | 55.1 | 110 | 73 | 4,620 | 208,827 | 7 | 95 | 0 |
| EC010 | NXIM00000000 | 61 | M | Tshwane Academic | + | E. hormaechei | 757 | 5.0 | 55.1 | 107 | 73 | 4,763 | 203,535 | 8 | 98 | 0 |
| EC015 | NXJF00000000 | 18 | M | Tshwane Academic | + | E. hormaechei | 113 | 4.9 | 50.6 | 72 | 56 | 4,604 | 344,169 | 6 | 96 | 0 |
| K001 | NXJG00000000 | 32 | F | Tshwane Academic | + | K. variicola | 403 | 6.0 | 57.0 | 75 | 56 | 5,913 | 105,256 | 18 | 90 | 1 |
| K006 | NXJH00000000 | 27 | F | Tshwane Academic | + | E. hormaechei | 260 | 5.0 | 56.0 | 79 | 68 | 4,824 | 861,27 | 37 | 92 | 0 |
| K130 | NXKO00000000 | | | | + | E. hormaechei | 109 | 5.1 | 54.8 | 74 | 59 | 4,801 | 190,037 | 9 | 90 | 0 |
| K063 | NXJN00000000 | 41 | M | Tshwane Academic | + | E. hormaechei | 223 | 5.2 | 55.9 | 102 | 78 | 4,951 | 60,240 | 55 | 90 | 2 |
| PM005 | NXKC00000000 | 29 | M | Tshwane Academic | + | P. alcalifaciens | 96 | 4.0 | 39.2 | 72 | 60 | 3,542 | 193,947 | 8 | 102 | 0 |

enem resistance through hyperexpression of ESBLs/AmpCs, porin downregulation, and carbapenemase production (2–4).

Subsequently, colistin was reintroduced into clinical medicine as a last-resort agent to manage infections resistant to carbapenems, other $\beta$-lactams, and other antibiotics (5, 6).

Inexorably, the increased use of colistin also resulted in the emergence of colistin resistance (3, 5, 7, 8). Since 2016, the discovery of a mobile colistin resistance gene, *mcr-1*, has enhanced the dissemination of colistin resistance worldwide (5, 9, 10). Several alleles of the *mcr-1* gene, such as *mcr-1* to *mcr-9*, have been also discovered, mainly in Europe and China in several *Enterobacteriaceae* species, including *Escherichia coli*, *Enterobacter* spp. *Klebsiella* spp., and *Salmonella* spp., etc. (11–14). Worryingly, increasing reports of isolates coharboring carbapenemases, ESBLs, and *mcr* genes are being reported worldwide and on self-transmissible plasmids (4, 15–19). Such strains may be pandrug resistant, restricting therapeutic options for clinicians and threatening public health (6, 8).

Although the WHO has classified ESBL- and carbapenemase-positive *Enterobacteriaceae* as critical and high-priority pathogens, relatively less attention has been given to rarely isolated pathogenic species such as *Citrobacter freundii*, *Enterobacter hormaechei*, *Klebsiella variicola*, *Providencia* spp., and *Proteus* spp. However, these relatively less isolated species can harbor important resistance genes that can be transferred to commonly isolated species such as *Klebsiella pneumoniae*, *Escherichia coli*, and *Salmonella enterica* (4, 11, 20–22). Hence, we investigated the resistome, mobilome, and genomic epidemiology of such less-isolated species to increase our understanding of their resistance mechanisms and epidemiology.

In this report, we describe for the first time the emergence of the *mcr-9.1* colistin resistance gene in South Africa and Africa as well as the cooccurrence of *mcr-9.1* with ESBLs in *Enterobacter hormaechei*. We further undertake a global phylogenomic analysis of *C. freundii*, *Enterobacter hormaechei*, *Klebsiella variicola*, and *Providencia/Proteus* spp., along with a global evolutionary analysis of *mcr-9.1*. These findings call for advanced One Health genomic surveillance to contain such multidrug-resistant strains from further dissemination.

## RESULTS

**Demographics, genome characteristics, and resistance phenome.** The specimens were mainly sourced from six males and four females, aged between 18 and 61 years and hospitalized at a single tertiary academic hospital in Pretoria, South Africa (Table 1). Except for the carbapenems, cephalothin, and combinations of clavulanate and tazobactam with cefotaxime/ceftazidime and piperacillin, respectively, most of the isolates were resistant to all the $\beta$-lactams. Moreover, reduced resistance to piperacillin-tazobactam (in isolates EC009, EC010, and K001) and sensitivity to aztreonam and cefepime (CF004) and cefoxitin (EC015, K001, K006, K130, K063, and PM005) were observed. All the isolates were phenotypically positive for ESBL production, except for CF004 (see Table S1 in the supplemental material; Table 1). None of the isolates was

**TABLE 2** Point mutations on chromosomal genes conferring colistin resistance in the *E. hormaechei* isolates from South Africa

| Isolate[a] | MIC (µg/ml) | Mutation(s) in: | | | | |
|---|---|---|---|---|---|---|
| | | pmrB | pmrA | phoP | phoQ | mgrB |
| EC001 | ≤2 | S175N, T210S, I227L, A233T, A344T | None | None | None | M1V |
| EC009 | ≤2 | | | | | |
| EC010 | >4 | | | | | |
| EC015 | >4 | | | | | |
| K006 | ≤2 | T121A, I134V, A344T | | | | |
| K063 | >4 | | | | | |
| K130 | ≤2 | | | | | |
| PM005 | >4 | —[b] | — | — | — | — |

[a]The reference genome used was *Enterobacter hormaechei* strain C15117 (PRJNA494598).
[b]—, intrinsic resistance.

resistant to amikacin. CF004 was susceptible to all the non-β-lactam antibiotics, while the remaining strains were resistant to almost all the non-β-lactam antibiotics except tigecycline and fosfomycin. Four of the isolates were resistant to colistin (Table S1).

The draft genome sizes of the genomic sequences ranged from 4 Mb to 6 Mb, with PM005 having the lowest GC content (39.2%) and number of coding sequences (3,542). The genome coverage ranged from 90× to 102×, while CRISPR arrays were identified in only two isolates (K001 and K063) (Table 1; Data Set S1).

The isolates were all multidrug resistant, and agreements as well as discrepancies between the resistance phenotypes (phenomes) and resistance genes (resistomes) were observed (Table S1). In particular, CF003 and CF004 had only *bla*$_{CMY}$ and *aac(6′)-If* or *fosA7* genes but expressed multidrug resistance to several antibiotics for which no resistance genes were identified. Such observations were also made with the presence of *fosA* genes in the isolates and the absence of fosfomycin resistance in almost all the strains as well as the presence of chromosomal AmpCs, such as ACT, LEN, and CMY in EC015, K001, K006, K130, K063, and PM005, which were all susceptible to cefoxitin (Table S1). Notable was the absence of colistin resistance in K006 and K130, which had the *mcr-9.1* gene; comparative genomic analyses showed that the colistin-susceptible and -resistant strains shared the same chromosomal mutations in *mgrB* and *pmrB* (Table 2). The absence of carbapenem resistance was corroborated by the absence of any carbapenemase gene, while the ESBL-positive phenotypes agreed with the ESBL genes identified in the genomes (Table S1; see Fig. 2).

**Resistome, mobilome, and evolutionary epidemiology of *mcr-9*.** CF003 and CF004 had the least number of resistance genes, with no ESBL genes. Dominant ESBLs in the other nine isolates included CTX-M-15, TEM-1, OXA-1, OXA-9, and ACT (Table S1 and Data Set S1). The genetic environment of the contigs on which the AmpC genes such as *bla*$_{CMY}$, *bla*$_{ACT}$, *bla*$_{LEN}$, and *bla*$_{LAP}$ were located strongly suggested their presence on chromosomes (Fig. 1A); this was confirmed by subjecting the contigs to a BLAST search on GenBank, and the sequence percent identities of the AmpC contigs aligned closely with only chromosomes. Notably, the *bla*$_{CMY}$ in PM005 was in close synteny with IS*Ec9* (Fig. 1A, panel III). *bla*$_{CTX-M-15}$ and *bla*$_{TEM-1}$ ESBLs were almost always found on the same contig, bracketed by IS*Ec9* and a Tn*3* transposon and IS*19* and an integrase/recombinase, respectively. In particular, *aph(6′)-Id::aph(3′′)::sul2* resistance genes were commonly found joined to IS*19*, just after *bla*$_{TEM-1}$ in *mcr-9*-negative *E. hormaechei* (Fig. 1A). In PM005, K006, and K130, the genetic environment of *bla*$_{TEM-1}$ was different (Fig. 1A). The *bla*$_{OXA}$ gene was commonly found between *catB3* and *aac(6′)-Ib* on a class 1 integron as gene cassettes or directly on the chromosome, similar to what was observed for *bla*$_{SHV}$, *bla*$_{LEN}$, and *bla*$_{ACT}$.(Fig. 1A).

A recent allele of the mobile colistin resistance gene, *mcr-9.1* (Fig. 1B), was identified in three *E. hormaechei* strains, the first to be discovered from Africa, dating back to 2013. *mcr-9.1* genes were all found in the immediate environment of a cupin fold metalloprotein, WbuC, similar to the *bla*$_{CTX-M-15}$ gene. *mcr9.1* only occurred once with

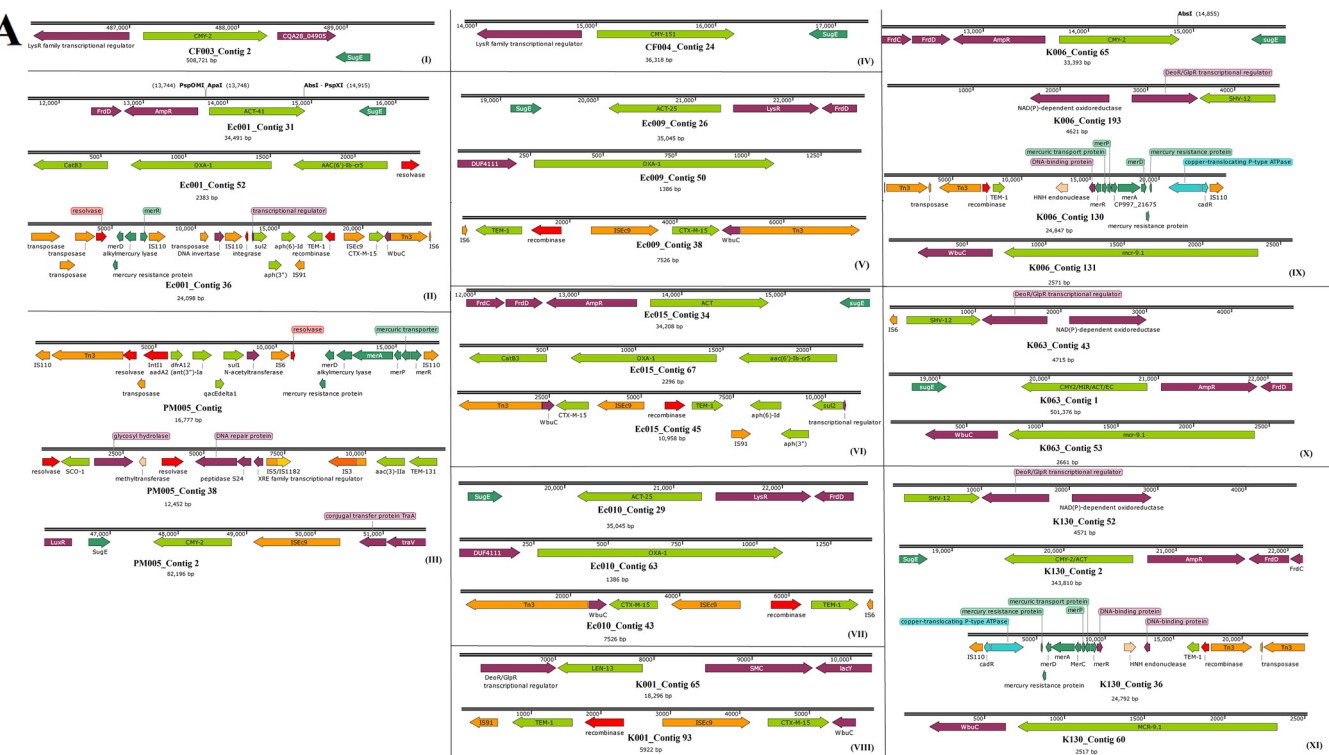

**FIG 1** Genetic environment of β-lactamase and *mcr-9.1* genes and global evolution of *mcr* genes. (A) The mobile genetic elements, namely, integrons/recombinases (red arrows) and transposons/ISs (orange arrows) bracketing the resistance genes (green arrows) are shown for the various isolates in panels I to XI. (B) Evolutionary relationships between the various *mcr* genes. *mcr-1* genes are indicated in red, *mcr-2* genes are in blue, *mcr-3* genes are in green, *mcr-5* genes are in sea blue, *mcr-6* is in gray, *mcr-7* is in orange, *mcr-8* is in turquoise, *mcr-9* is in black, and *mcr-10* is in violet/pink. The clustering of the various *mcr* alleles and their evolutionary distance from each other reflect the sequence similarity and differences between the various alleles. Bootstrap values for panel B are shown on the branches of Fig. S1 in the supplemental material.

*bla*$_{CTX-M-15}$ in K063 (Fig. 1A). A nucleotide BLAST search of the contigs, namely, K006 contig 131, K063 contig 53, and K130 contig 60, and subsequent phylogenetics analysis using the fast minimum evolution method showed that the contigs harboring the *mcr9.1* gene were most closely aligned to plasmid genomes from different *Enterobacteriaceae* species, with a few chromosomes such as that of *Enterobacter kobei* strain DSM 13645 being also closely aligned (Fig. 2). Sequence alignment showed close similarity between the three *mcr-9*-bearing contigs (Data Set S2), particularly between the contigs of K006 and K130 (Fig. 2A and B). Notably, both K006 and K130 aligned with very similar nucleotide identities with the same genomes, although minor differences existed in their nucleotide sequences (Data Set S2).

Both K006 and K130 were phylogenetically distant from K063 (Fig. 2A and B), which is further evinced by the different genomes to which they both most closely aligned and the distance of those genomes from each other in the distance trees (Fig. 2A to C). The evolutionary epidemiology of the *mcr-9.1*-bearing plasmids or chromosomes is further shown in Fig. 2A to C, with different plasmid types and chromosomes mediating the spread of this gene within and between species throughout the world. The various *mcr-9.1*-bearing genomes (plasmids or chromosomes) clustered phylogenetically into six clades, herein labeled A to F, with *Enterobacter* sp. strain 18A13 plasmid pECC18A13-1 DNA from Japan seeming to be the earliest ancestor and *Enterobacter hormaechei* strain S13 plasmid pSHV12-1301491 from the United States seeming to be the most recent. All the *mcr-9.1* contigs in this study fell within phylogenetic cluster B, although the genomes that aligned most closely with these contigs changed their phylogenetic clustering between K006/K130 and K063 (Fig. 2A to C).

Several integrons bearing gene cassettes of antibiotic resistance genes were identified in various contigs in the isolates (Table 3), with none being detected in the *C.*

**B**

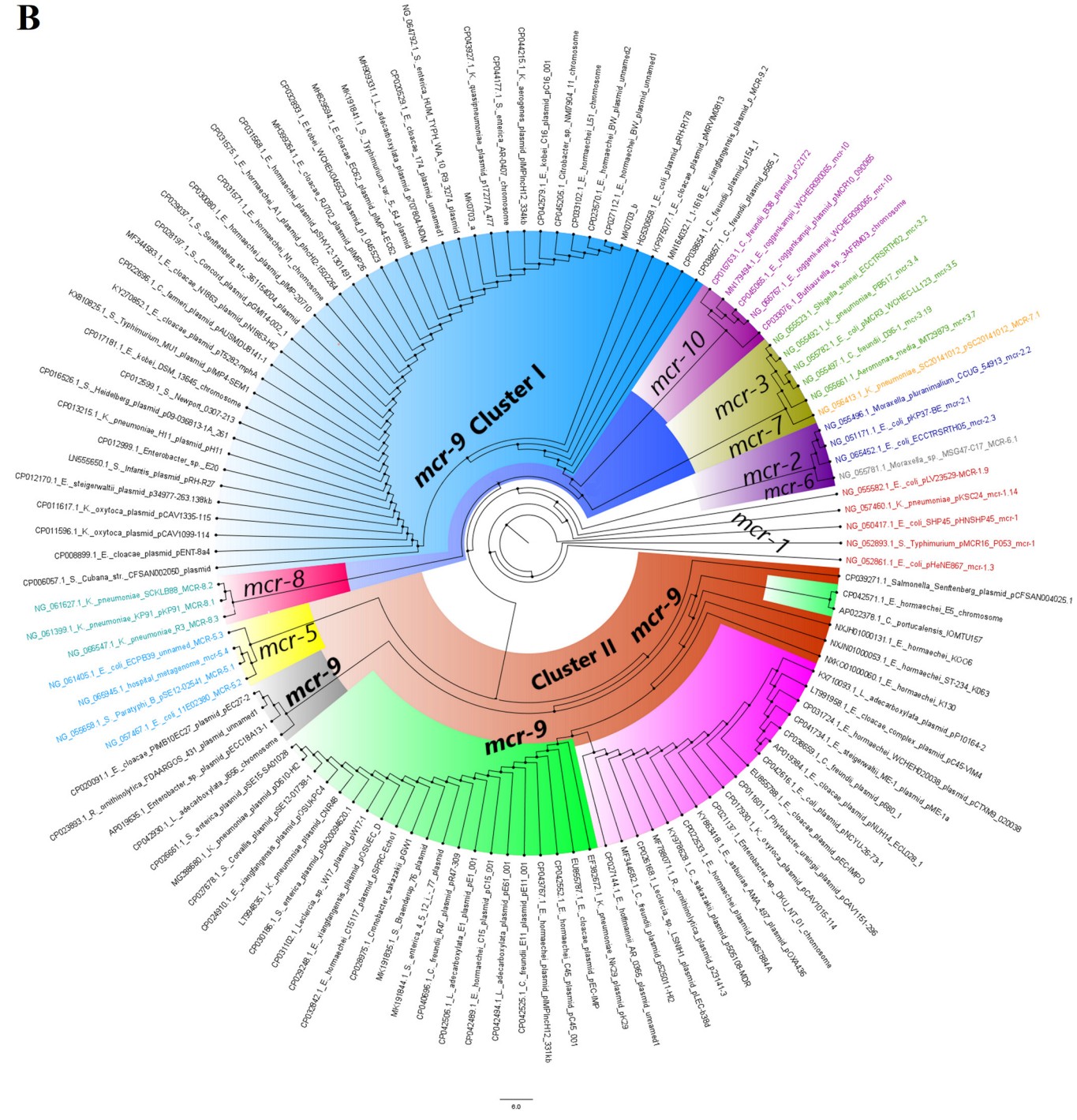

**FIG 1** (Continued)

*freundii* strains. PM005 (*Providencia alcalifaciens*) had a unique integron, In*27*, while K001 (*K. variicola*) and four *E. hormaechei* strains harbored In*191*. K006, K063, and K130, originating from different patients and having the *mcr-9.1* gene, harbored the same integrons and very similar gene cassettes. EC009 and EC010 had the same class 1 integron and gene cassettes and belonged to the same clone (Table 3). Most of the isolates had multiple plasmid replicons, with ColRNAI being the most common; ColRNAI was also found in the *mcr-9.1*-positive strains. The isolates were further grouped using their plasmid multilocus sequence types (pMLST). Except for CF004 and PM005, which

**A**

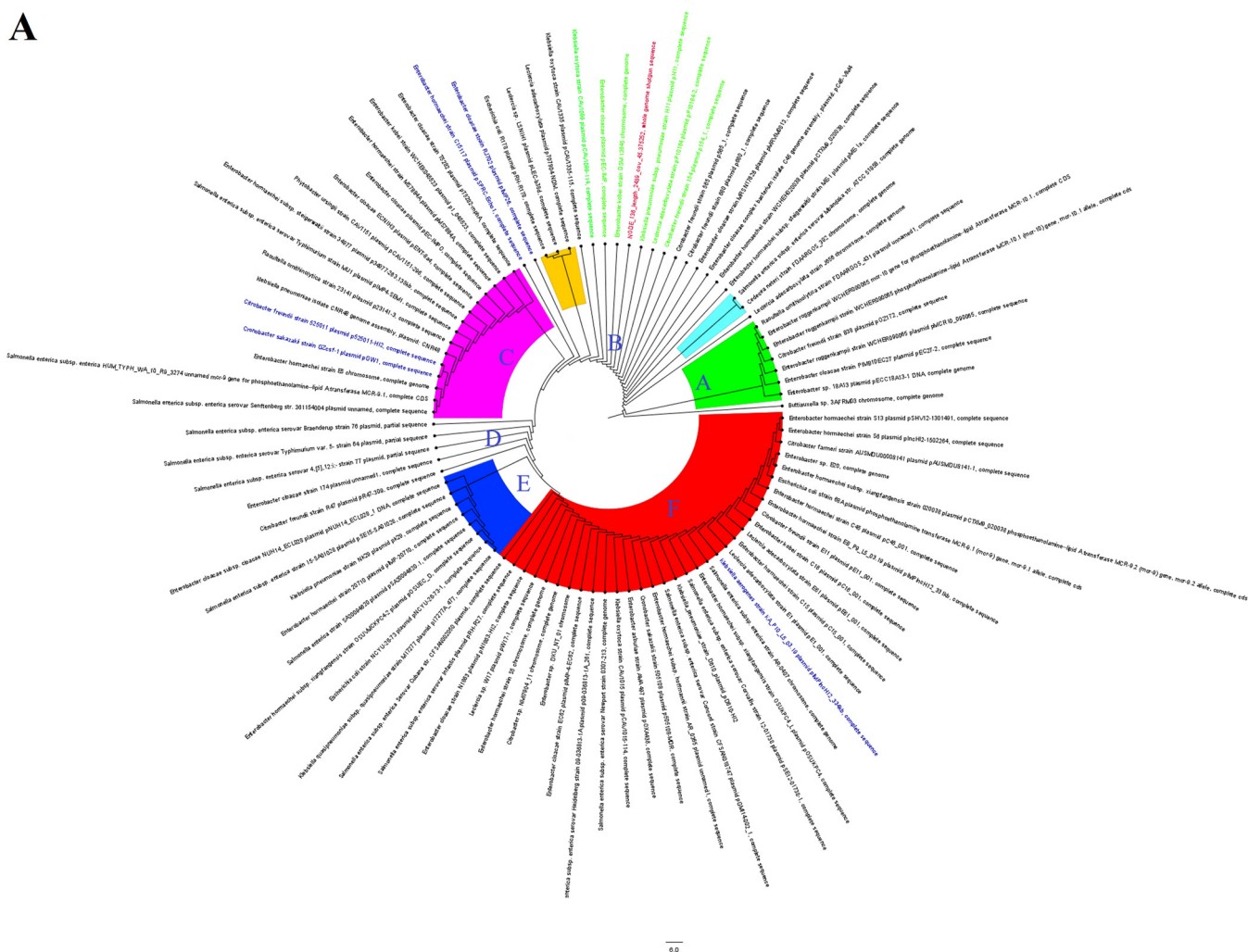

**FIG 2** Evolutionary epidemiology of *mcr-9.1*-harboring strains. The evolutionary distance and relationship between the contigs bearing *mcr-9.1* and highly similar genomes obtained from GenBank are shown in three different trees for K006 (A), K130 (B), and K063 (C). The contigs from this study are indicated in red-colored text labels, and the genomes of closest alignment are indicated either in green-colored (for K006 [A] and K130 [B]) or blue-colored (for K063 [C]) text labels. The various phyletic groupings that form a separate clade are labeled from A to F with different colors showing the evolutionary trajectory of *mcr-9.1*-bearing strains/plasmids.

had no pMLST but contained an A/C$_2$ replicon, IncHI2[ST-1] and IncF subtypes were the main plasmid types. Notably, all the *mcr-9.1* strains had an IncHI2[ST-1] plasmid, while two, K006 and K063, coharbored IncF subtype plasmids. Although the resistance genes within strains having the same plasmid types were different, core resistance genes with same genetic environment were present (Fig. 1A; Table 4).

**Evolution of *mcr* genes.** The phylogenetic and evolutionary relationships between the 10 *mcr* variants are shown in Fig. 1B (see Fig. S1 in the supplemental material), with the *mcr-9* genes being clustered into two clusters, I and II. Cluster I of *mcr-9* was closely related to that of *mcr-10*, *mcr-3*, *mcr-7*, and *mcr-8*, while cluster II was closely related to that of *mcr-5*. Notably, strains containing *mcr-1*, one of which was used as the reference strain, were closely related to those containing *mcr-2* and *mcr-6*, with *mcr-2* and *mcr-6* clustering closely together. Particularly, there were more subclades within cluster II than cluster I of the *mcr-9* genes, which is due to the sequence differences between the genes in cluster II.

**Phylogenomics.** The *C. freundii* strains were not related to any South African or African strains but were related to American and European strains. CF004 and CF003 were not related to each other, but CF004 was of the same clone as *C. freundii*

## B

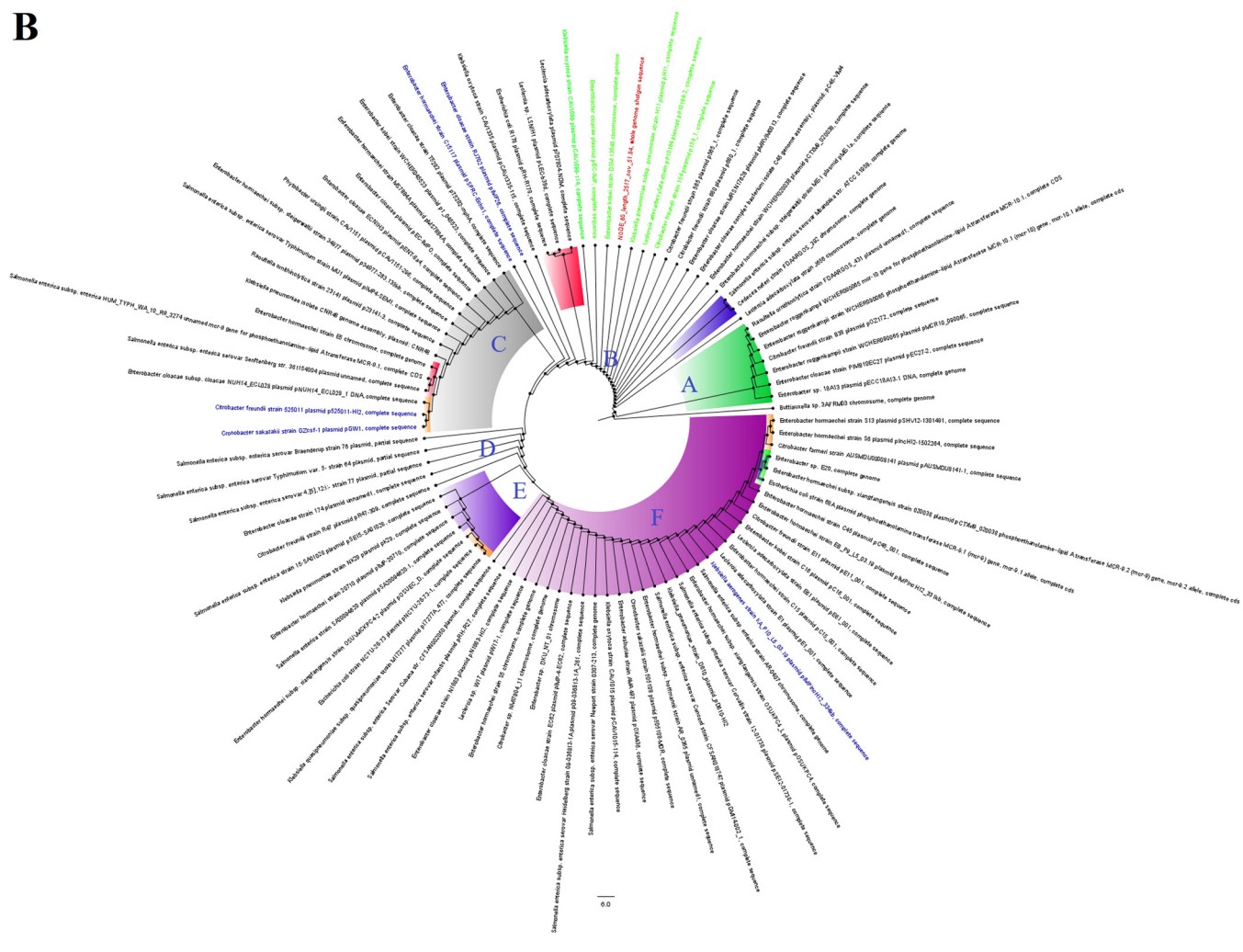

**FIG 2** (Continued)

CF_12_ST_92 [*aac(3)-IIa, aac(6')-Ib4, aadA1, aph(6)-Id, arr-2, bla*$_{CMY-48}$, *bla*$_{OXA}$, *bla*$_{VIM-1}$, *catB2, dfrA14, dfrB1, mph*(A), *qacE, qnrA1 sul1*] from Spain and of the same clade as strains 705SK3 (*bla*$_{CMY-75}$, *bla*$_{OXA-48}$) from Switzerland and CF_13_ST_93 [*aac(6')-Ib4, aadA1, aph(3'')-Ib, aph(6)-Id, bla*$_{CMY-75}$, *bla*$_{VIM-1}$, *catB2, dfrA14, dfrB1, qacEΔ1, qnrB10, sul1, sul2*] from Spain. CF003 was of the same clade as MGH142 (*bla*$_{CMY}$), UMH16 (*bla*$_{CMY}$), MGH141 [*aadA2, ant(2'')-Ia, aph(3')-II, bla*$_{CARB-2}$, *bla*$_{CMY}$, *bla*$_{KPC-3}$, *ble, catB11, catB3, cmlA1, dfrA19, mph*(E), *msr*(E), *qacE*Δ1, sul1], and CRE20 (*bla*$_{CMY}$) from the United States (Fig. 3).

The *E. hormaechei* strains were clustered into three clusters, with the *mcr-9.1* strains being very closely related within the same subclade; K006 and K063 were of the same clone. Moreover, EC001 and EC015 were of the same clone as EC009 and EC010 (Fig. 4A). However, EC001 and EC015 were distantly related in the presence of more closely related genomes (Fig. 4B and C). The *mcr-9.1*-positive strains were of close evolutionary relationship with other *E. hormaechei* strains from China, United Kingdom, United States, and Thailand: EB_P9_L5_03.19 [*aac(6')-Ib4, aph(3'')-Ib, aph(6)-Id, bla*$_{ACT-56}$, *bla*$_{IMP-1}$, *bla*$_{TEM-1}$, *catA, catA2, dfrA19, fosA, mcr-9.1, qacE, qacEΔ1, qnrA1, sul1*], 2011WA-NCV (*bla*$_{ACT-56}$, *catA, fosA*), 2011WA-SCV (*bla*$_{ACT-56}$, *catA, fosA*), 35415 (*bla*$_{ACT-43}$, *catA, fosA, oqxB*), WCHEH090020 (*bla*$_{ACT-56}$, *catA, fosA*), UBA4405, WCHEH090006 (*bla*$_{ACT-56}$, *catA, fosA*), T38-C141 [*aph(3'')-Ib, aph(6)-Id, bla*$_{ACT-17}$, *bla*$_{CMY}$, *bla*$_{OXA-48}$, *bleO, catA, dfrA14, fosA, mcr-1.1, mcr-3.1, sul2, tet*(A)], UBA6755, S6 [*aac(6')-II, aadA1, aadA2, aph(3'')-*

C

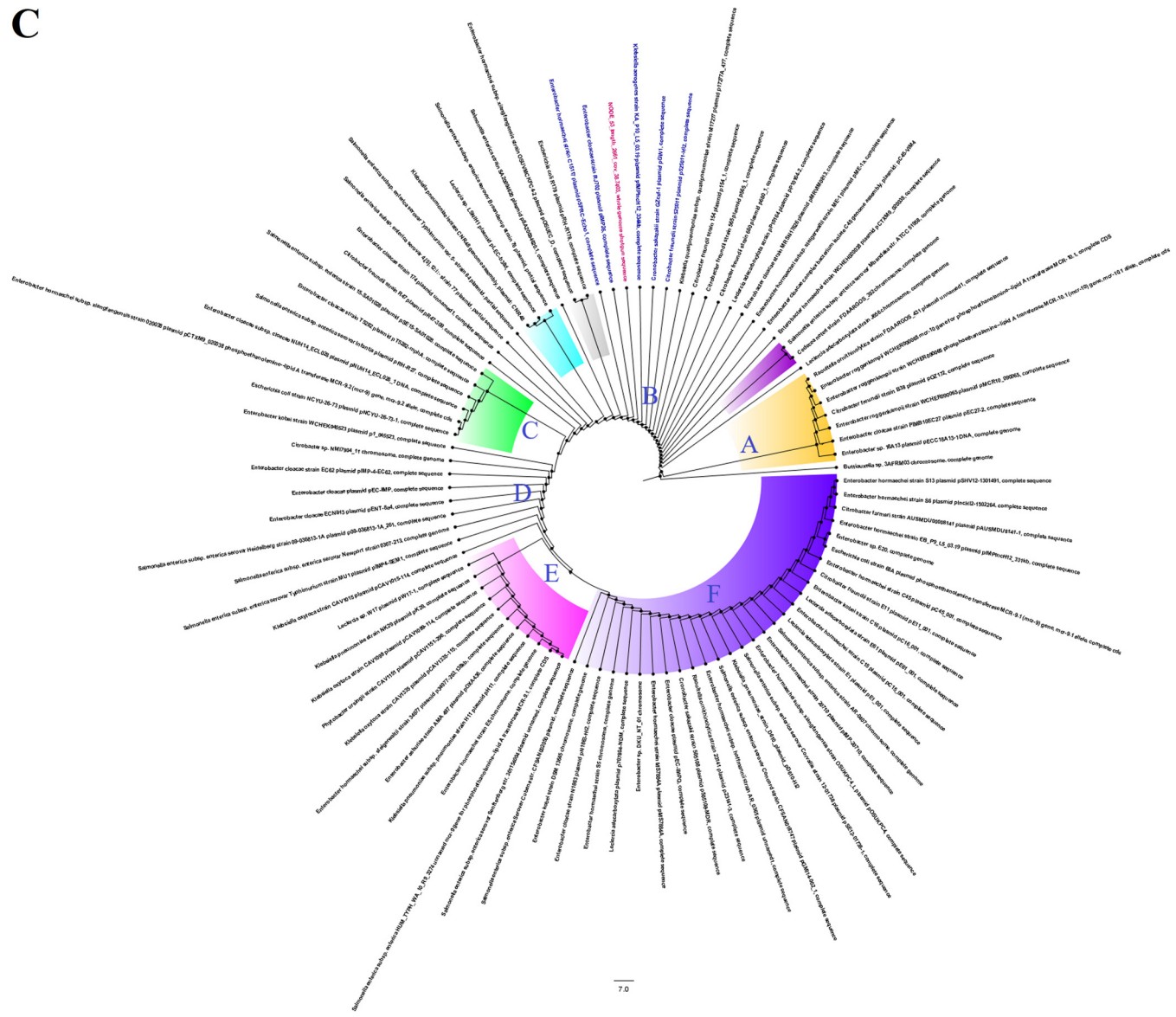

**FIG 2** (Continued)

*lb, aph(6)-Id, bla*~ACT-27~, *bla*~SHV-12~, *bla*~TEM-1~, *bla*~VIM-1~, *catA, dfrA1, dfrA12, fosA, mcr-9.1, mph*(A), *oqxB, qacE, qacE*Δ*1, qnrA1, sul1*], and UBA5648 (Fig. 4A and B).

The closely related EC009 and EC010 strains were of close evolutionary distance to Chinese and American strains, i.e., WCHEH085055 [*aac(3)-IId, aadA5, bla*~ACT-41~, *bla*~IMP-4~, *bla*~TEM-1~, *catA, catA2, dfrA17, fosA, mph*(A), *oqxB, qacE*Δ*1, qnrS1, sul1, tet*(D)], UBA7898, and GN04900 (*bla*~ACT-41~, *catA, fosA, oqxB*). EC001 and EC015 were of a close phyletic group with strains from the United States, China, Croatia, Spain, and Turkey. For EC001, UBA6646, SMART_488 [*aac(6')-Ib-cr5, aph(3'')-Ib, aph(3')-Ia, aph(3')-VIa, aph(6)-Id, bla*~ACT~, *bla*~CTX-M-15~, *bla*~KPC-2~, *bla*~OXA-1~, *bla*~TEM-1~, *catA, catA2, catB3, dfrA14, fosA, oqxB, qnrB19, sul2, tet*(A)], SMART_723 [*aac(3)-II, aac(6')-IIc, aph(3'')-Ib, aph(6)-Id, arr, bla*~ACT-45~, *bla*~DHA-7~, *bla*~OXA-48~, *bla*~SHV-12~, *bla*~TEM-1~, *catA, catA2, dfrA19, ere*(A), *fosA, mcr-9.1, oqxB, qacE, qnrB4, sul1*], and WCHEX090075 [*aadA2, ant(2'')-Ia, bla*~ACT~, *bla*~NDM-1~, *bla*~SHV-12~, *ble, catA, catA1, dfrA14, fosA, mcr-9, oqxB, qacE, qnrA1, qnrS1, sul1*] were of the same clade, while WCHEH090043 (*bla*~ACT~, *catA, fosA, oqxB*), UBA1647, SMART_530 (*bla*~ACT~, *bla*~OXA-48~, *catA, fosA, oqxB, qnrS1*), SMART_1112 [*aac(6')-Ib4, aac(6')-Im, aadA1, aadA16, aph(2'')-IIa, aph(3'')-Ib, aph(3')-Ia, aph(6)-Id, arr-3, bla*~ACT~, *bla*~CTX-M-15~, *bla*~TEM-1~, *bla*~VIM-1~, *catA, catA2,*

**TABLE 3** Gene cassettes and integrons found in the isolates

| Sample code (MLST) | Integron(s) | Cassette arrays (location on contig)[a] |
|---|---|---|
| CF003 | None | None |
| CF004 | None | None |
| EC001 (ST-459) | In191, In705, In363 | *dfrA14b*$_{(94-660)}$-*attC*$_{(581-666)}$ |
| EC009 (ST-231) | In191 | *dfrA14b*$_{(593-1159)}$-*attC*$_{(587-672)}$; *arr3*$_{(1584-2186)}$-*attC*$_{(1578-1691)}$-*ereA3*$_{(120-1583)}$-*attC*$_{(114-170)}$-*aadA1*$_{(1-119)}$-partial |
| EC010 (ST-231) | In191 | *dfrA14b*$_{(471-1037)}$-*attC*$_{(465-550)}$; *arr3*$_{(1584-2186)}$-*attC*$_{(1578-1691)}$-*ereA3*$_{(120-1583)}$-*attC*$_{(114-170)}$-*aadA1*$_{(1-119)}$ |
| EC015 | In191, In705 | *aadA1ai*$_{(52-907)}$-*attC*$_{(854-913)}$-3'CS;*dfrA14b*$_{(5384-5950)}$-*attC*$_{(5378-5463)}$) |
| K001 (ST-1791) | In191, In792 | *dfrA14b*$_{(53-619)}$-*attC*$_{(540-625)}$; *qacEΔ1*$_{(1484-1584)}$-*aacA4*$_{(815-1483)}$-*attC*$_{(814-880)}$-*arr3*$_{(212-819)}$-*attC*$_{(206-319)}$-3'CS |
| K006 | In46, In127, In615 | *aacA4*$_{(2592-3230)}$-*attC*$_{(2586-2657)}$; *aadA2*$_{(80-935)}$-*attC*$_{(74-133)}$-3'CS; *aacA27*$_{(5380-6117)}$- *attC*$_{(5374-5443)}$-*ereA2Δ*:: IS1247$_{(3446-5116)}$, *aac3*$_{(2616-3425)}$ *arr7*$_{(2075-2470)}$-*attC*$_{(74-130)}$ |
| K130 (ST-90) | In46, In127, In615 | *aadA2*$_{(53-908)}$-*attC*$_{(47-106)}$-3'CS; *aacA4*$_{(2566-3204)}$-*attC*$_{(2560-2631)}$; *aacA27*$_{(5352-6089)}$-*attC*$_{(5346-5415)}$-*ereA2D*:: IS1247$_{(3419-5088)}$, *aac3*$_{(2589-3398)}$*arr7*$_{(2048-2443)}$-*attC*$_{(47-103)}$ |
| K063 | In46, In127, In615 | *aadA2*$_{(125-980)}$-*attC*$_{(119-178)}$-*qacEΔ1*-3'CS;*aacA4*$_{(2638-3276)}$-*attC*$_{(2632-2703)}$; *aacA27*$_{(5425-6162)}$-*attC*$_{(5419-5488)}$-*ereA2D*, *aac3*$_{(2661-3470)}$, *arr7*$_{(2120-2515)}$-*attC*$_{(119-175)}$-*qacEΔ1*-3'CS |
| PM005 | In27 | *dfrA12*$_{(5645-6228)}$-*attC*$_{(6145-6234)}$-*gcuF*$_{(6229-6548)}$-*attC*$_{(6495-6554)}$-*aadA2*$_{(6549-7404)}$-*attC*$_{(7351-7410)}$-*qacEΔ1*$_{(7513-7860)}$-*sul1*$_{(7854-8693)}$-orf5$_{(8821-9321)}$-3'CS |

[a]3'CS refers to the 3' end of the coding sequence (DNA).

*dfrA19, dfrA27, fosA, mcr-9.1, oqxB, qacEΔ1, qnrB6, sul1, sul2, tet*(A), *tet*(D)], SMART_562 [*aac(6')-Ib4, aac(6')-Im, aadA1, aadA16, aph(2'')-IIa, aph(3'')-Ib, aph(3')-Ia, aph(6)-Id, arr-3, bla*$_{ACT}$, *bla*$_{CTX-M-15}$, *bla*$_{TEM-1}$, *bla*$_{VIM-1}$, *catA, catA2, dfrA19, dfrA27, fosA, mcr-9.1, oqxB, qacEΔ1, qnrB6, sul1, sul2, tet*(A), *tet*(D)] etc., clustered together (Fig. 4).

The closest evolutionary relative of K001 was a single strain, WUSM_KV_44 (*bla*$_{LEN-13}$, *fosA, oqxA, oqxB*) from the United States (Fig. 5), while PM005, which was initially identified as *Proteus mirabilis* by MicroScan but later revised by NCBI's ANI to *P. alcalifaciens*, was not related to any *Providencia* sp. genome (Fig. 6A and B); however, it was closely related to *Proteus mirabilis* strain Pr2921 from Uruguay [*catA, tet*(J)] (Fig. 6C).

## DISCUSSION

We here report on the first emergence of *mcr-9.1* in both South Africa and Africa in three *E. hormaechei* strains with a rich repertoire of resistomes and mobilomes. The *mcr-9.1* strains were collected among clinical *Enterobacteriaceae* strains during a molecular surveillance procedure to identify ESBL producers in a referral laboratory. It is a worrying observation that all the strains, sourced from different patients and wards within the same hospital, were multidrug resistant. Although the resistomes largely reflected the resistance phenotype, resistance to antibiotics was observed without an underlying resistance gene being identified. Specifically, the *C. freundii* strains expressed resistance to several antibiotics, although not more than three resistance genes were found in them. This could suggest the presence of an unknown resistance determinant or the use of active efflux. Furthermore, sensitivity to certain antibiotics such as colistin was not corroborated by the presence of *mcr-9.1*. Sensitivity to colistin in strains having the *mcr* gene has been reported previously, and mutations in chromosomal colistin resistance genes have been shown to exert a higher resistance MIC than *mcr* genes (3, 23–25).

Only a single strain with the *mcr* gene was colistin resistant, and the identified mutations in *mgrB* and *pmrAB* were also found in susceptible strains. Hence, the identified mutations cannot be responsible for colistin resistance. Consequently, other factors identified for colistin resistance or other unknown determinants could be responsible for the observed colistin resistance (3, 7, 11, 22). The absence of colistin resistance in *mcr-9.1* strains is not new or surprising, as it has been observed in other studies (23, 24, 26). However, it should be known that in the presence of certain promoters in some bacterial species, these silent genes can become actively expressed, leading to phenotypic resistance (10, 23). Furthermore, it has been suggested that a reduction in the colistin breakpoints might increase the detection of otherwise colistin-

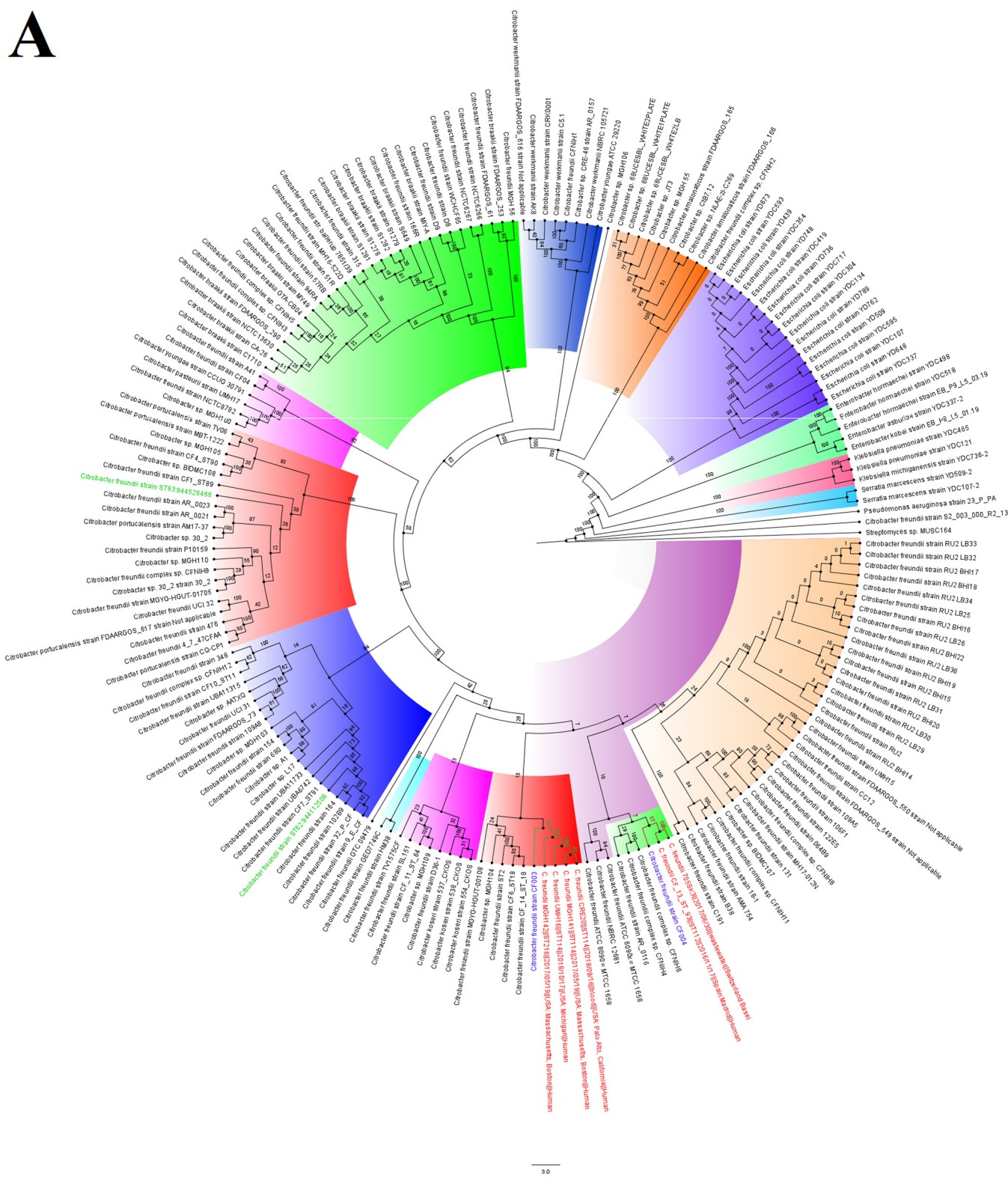

**FIG 3** Global phylogenomics of *Citrobacter freundii* strains obtained from PATRIC/GenBank (A and B). The relationships of the two *C. freundii* strains, CF003 and CF004 (shown in blue-colored text labels), to all *C. freundii* genomes deposited in GenBank and PATRIC were analyzed and drawn into two trees. (A) Genomes belonging to the same subclade with the closest evolutionary distance are indicated in red-colored text labels, while isolates from South Africa are indicated in green-colored text labels. (B) The various phylogenetic clades and subclades are highlighted together and uniquely to show their evolution and epidemiology. The trees were drawn using the maximum-likelihood method in RAxML, using *Streptomyces* sp. strain MUSC164 as a reference. Bootstrap values are shown on the branches.

# B

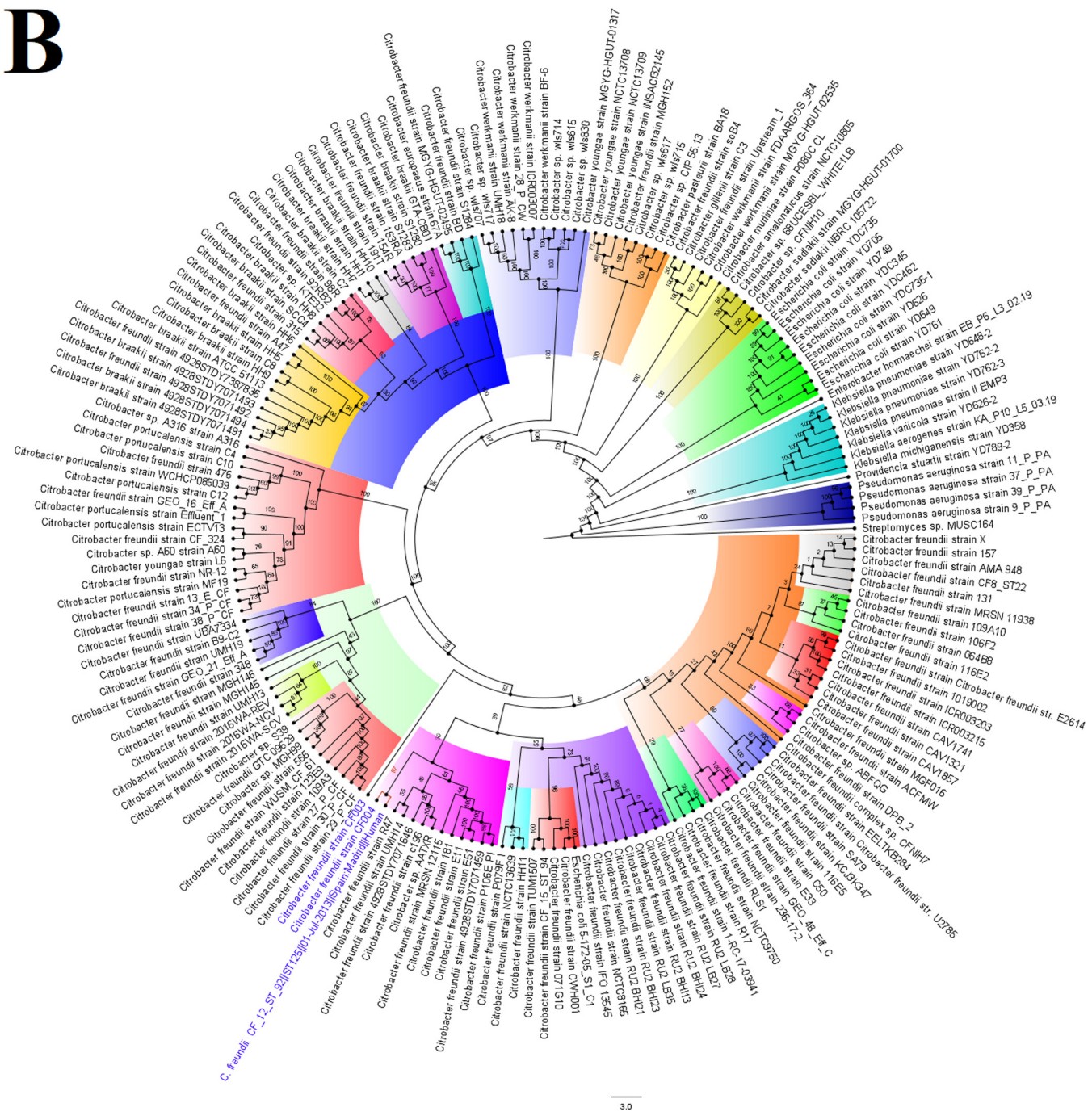

**FIG 3** (Continued)

resistant strains that are classified as susceptible, although this reduction must be done after additional investigations (26).

The three strains bearing the *mcr-9.1* gene were closely related, but the contigs bearing the *mcr-9.1* genes did not have 100% nucleotide sequence identity among themselves (see Data Set S2 in the supplemental material), with the *mcr-9.1* contigs' distance trees showing that K006 and K130 were of a more recent common ancestor (Fig. 2). It is worth noting that K006 and K130 were phylogenetically distant but that their *mcr-9.1* contigs were of closer nucleotide identity than those of K006 and K063, which were of the same clone (Fig. 4). This observation, plus the close alignment of the *mcr-9.1* contigs

mSystems®

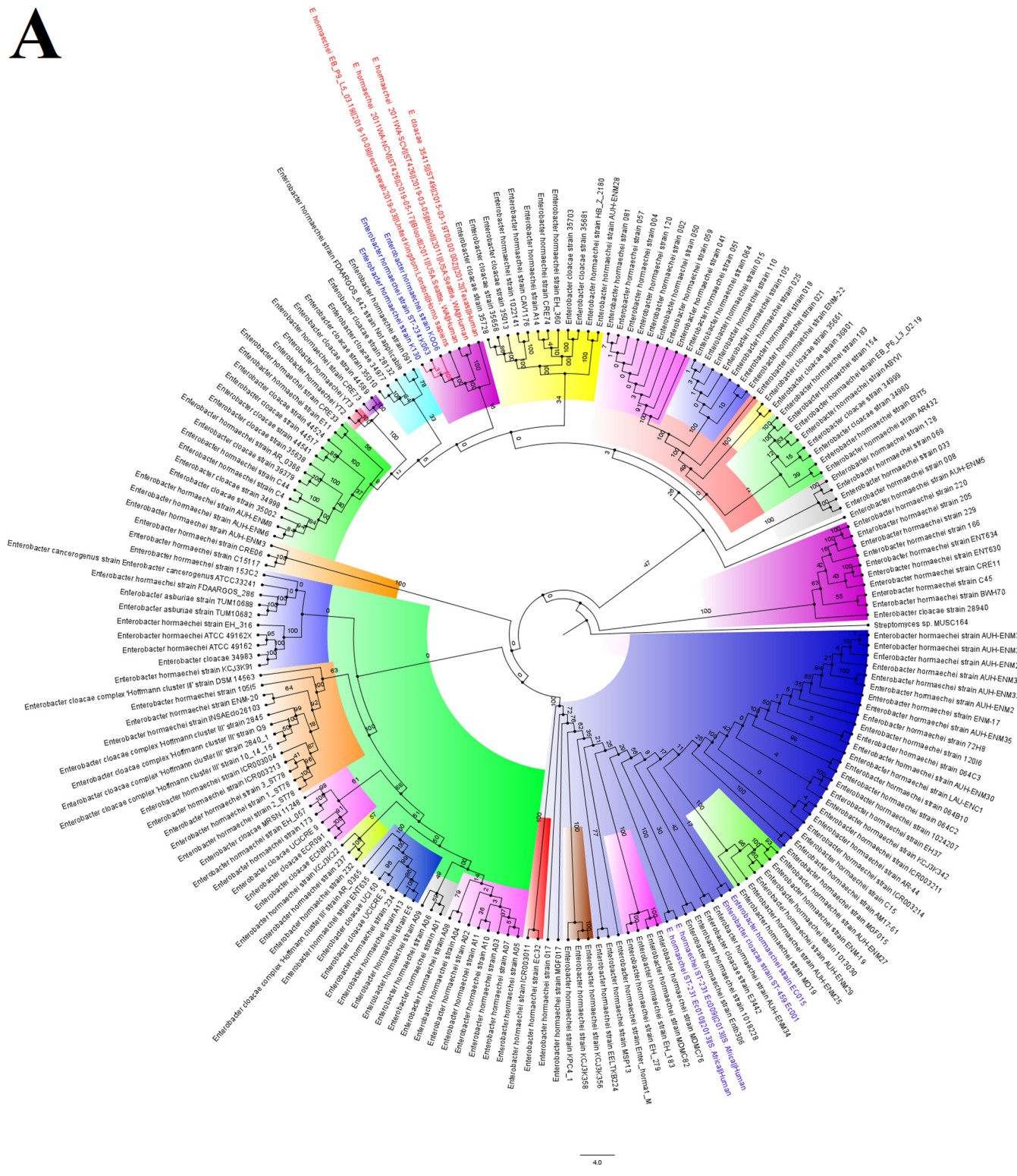

**FIG 4** Global phylogenomics of *Enterobacter hormaechei* strains obtained from PATRIC/GenBank. The relationships of the seven *E. hormaechei* strains, EC001, EC009, EC010, EC015, K006, K063, and K130 (indicated in blue-colored text labels), to all *E. hormaechei* genomes deposited in GenBank and PATRIC were analyzed and drawn into three trees, as shown in panels A, B, and C. Genomes belonging to the same subclade with the closest evolutionary distance are indicated in red-colored text labels. The various phylogenetic clades and subclades are highlighted together and uniquely to show their evolution and epidemiology. The trees were drawn using the maximum-likelihood method in RAxML, using *Streptomyces* sp. MUSC164 as a reference. Bootstrap values are shown on the branches.

# B

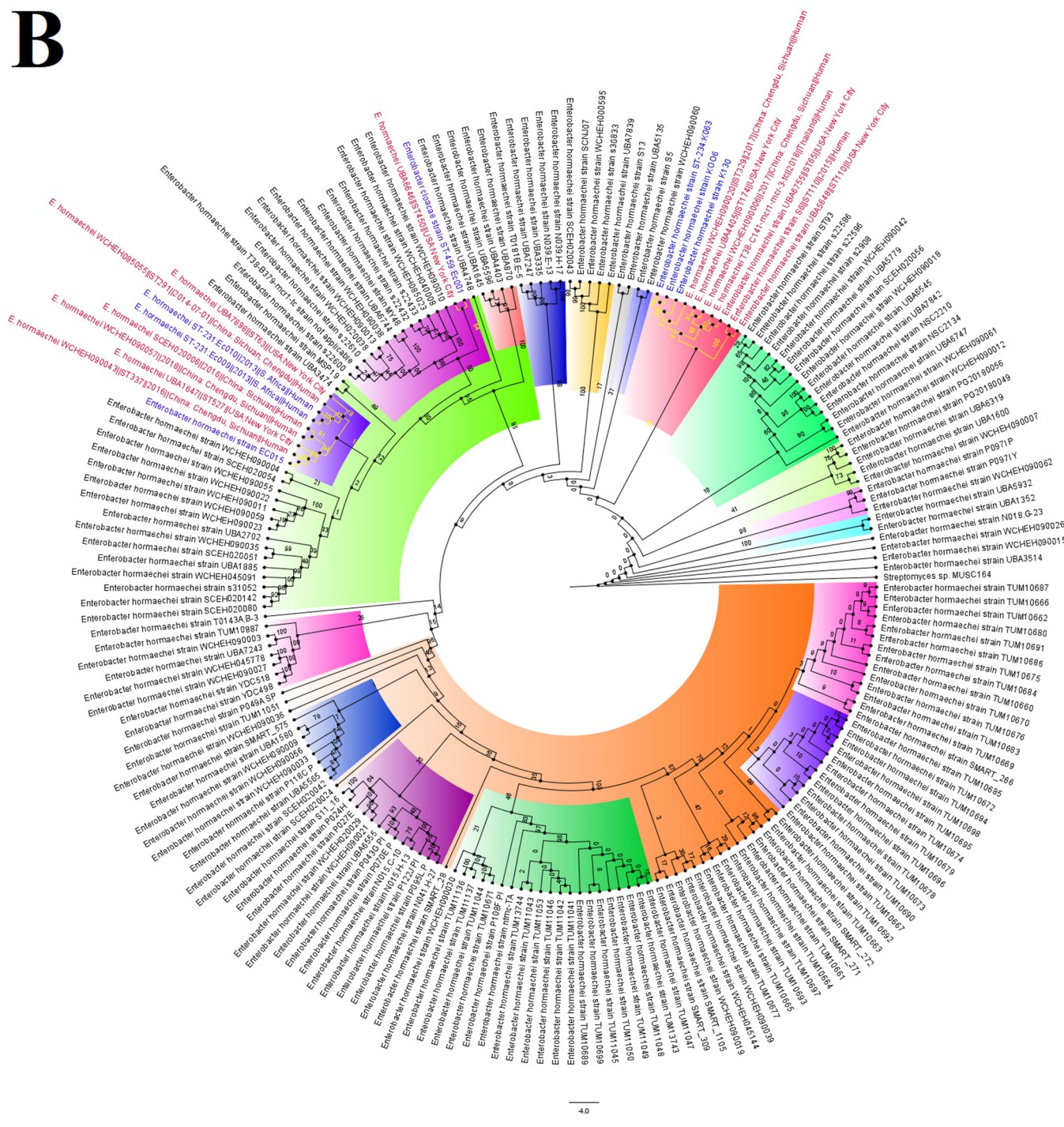

**FIG 4** (Continued)

with plasmid genomes in GenBank (Fig. 2), strongly suggests that the *mcr-9.1* gene might have been horizontally, instead of vertically, acquired. It also suggests that either the patients (one was female, and the other was male) from which K006 and K063 were obtained got the *mcr-9.1* gene from the same source or that one of the two patients transferred it to the other directly, through a health care worker, or via the hospital environment.

The clustering dynamics of the *mcr* genes confirm their nomenclature as *mcr* variants of the same allele clustered together on the same branch. For instance, all *mcr-1* strains clustered together, showing their close sequence identity (Data Set S3).

# C

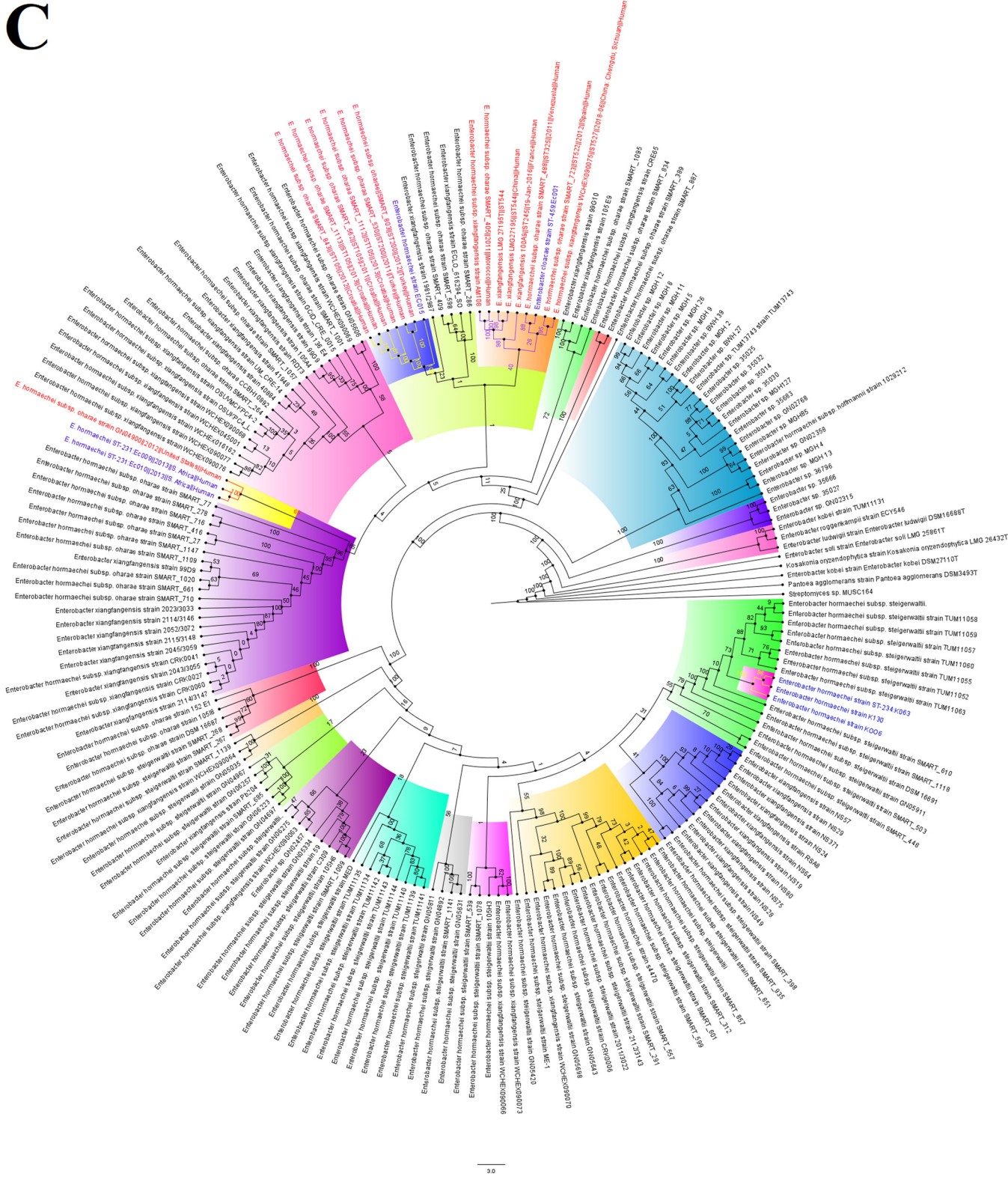

**FIG 4** (Continued)

The evolutionary distance between the various *mcr* variants also confirms their sequence differences stemming from the nucleotide polymorphisms between these alleles. These differences in sequence percent identity between the various *mcr* alleles begs the question of whether these variants evolved from a single ancestor or emerged

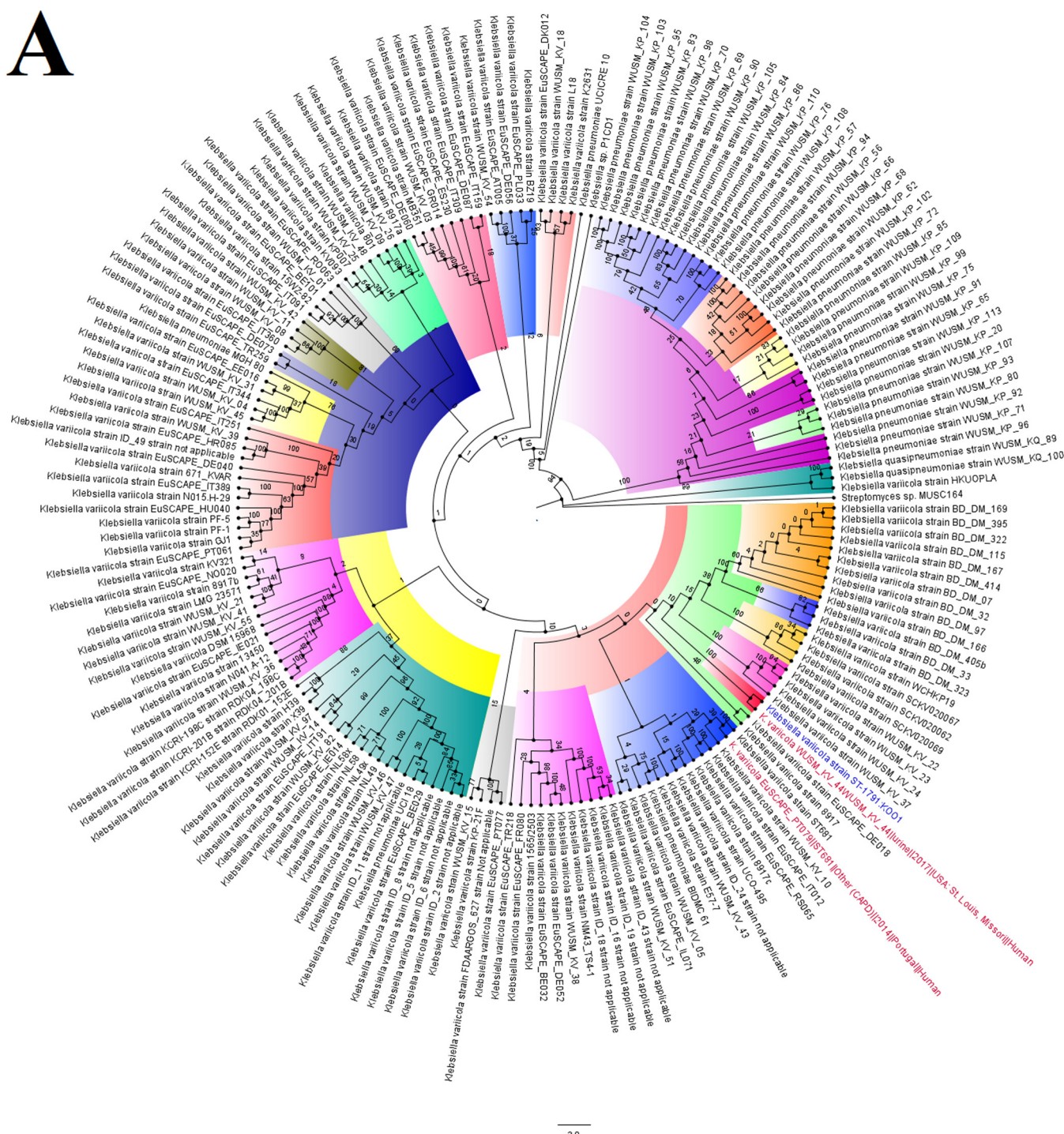

**FIG 5** Global phylogenomics of *Klebsiella variicola* strains obtained from PATRIC/GenBank. The relationship of *K. variicola* strain K001 (shown in blue-colored text labels) to all *K. variicola* genomes deposited in GenBank and PATRIC were analyzed and drawn into two trees, as shown in panels A and B. Genomes belonging to the same subclade with the closest evolutionary distance are indicated in red-colored text labels. The various phylogenetic clades and subclades are highlighted together and uniquely to show their evolution and epidemiology. The trees were drawn using the maximum-likelihood method in RAxML, using *Streptomyces* sp. MUSC164 as a reference. Bootstrap values are shown on the branches.

independently from each other. It can, however, be seen that some variants are from a more recent common ancestor than others, owing to their sequence percent identity (Fig. 1B; Fig. S1). The phenotypic effect of these variants, in terms of the level of colistin resistance, is yet to be determined to know which of these variants confers the most resistance in terms of MIC levels. Nevertheless, most of these *mcr* variants were plasmid

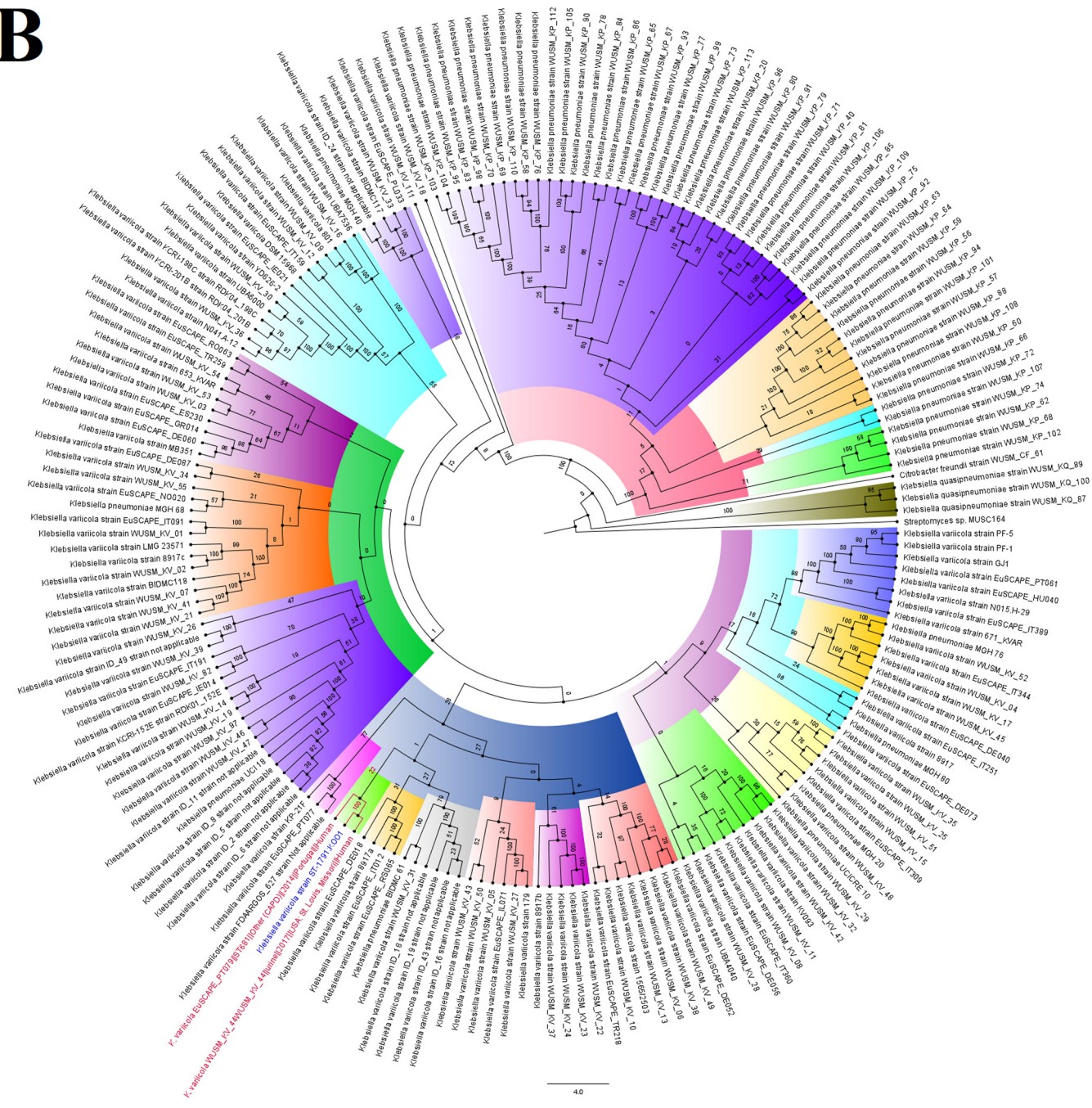

**FIG 5** (Continued)

borne, with a few being chromosomal, explaining their wide distribution in diverse species (11, 27).

The global evolutionary trajectory of the *mcr-9* gene and genomes shows six major phylogenetic clades, herein labeled A to F. The strains and plasmids in clade A seem to be the earliest ancestors of this gene, with clade F being the last and most recent. Moreover, *Enterobacter* spp., particularly *E. hormaechei*, and *Salmonella enterica* plasmids are the most common hosts of the *mcr-9* gene (Fig. 2). This is not surprising, as *mcr-9.1* was first identified in *Salmonella enterica* serovar Typhimurium (23). The diversity of species and plasmids involved in the dissemination of this gene explains its promiscuity and rapid spread around the globe, further corroborating the need to restrict the use of colistin in both veterinary and human medicine (5, 28).

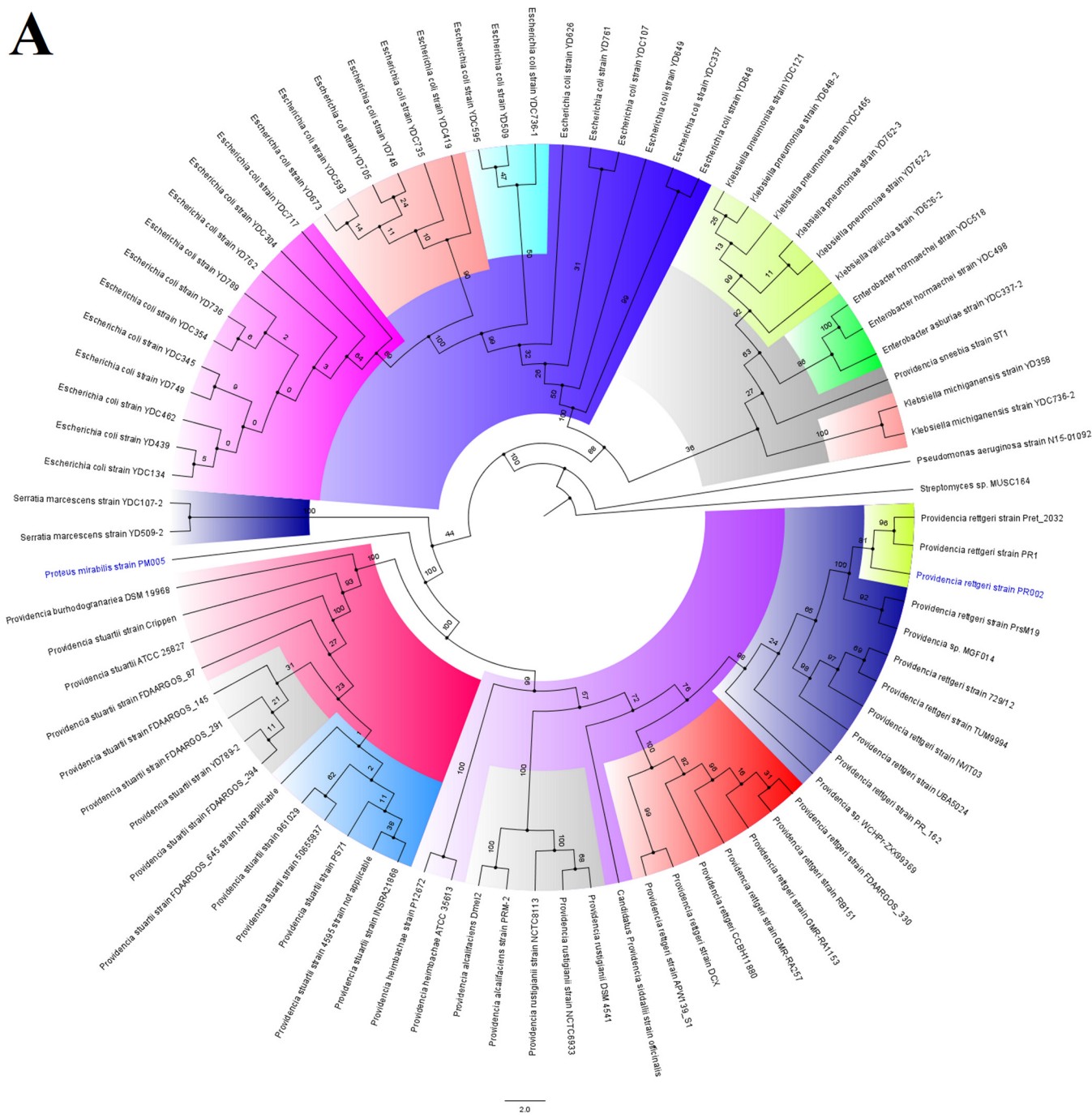

**FIG 6** Global phylogenomics of *Providencia alcalifaciens* and *Proteus mirabilis* strains obtained from PATRIC/GenBank. The relationship of *P. alcalifaciens* strain PM005 (indicated in blue-colored text labels) to all *Providencia* (A and B) and *Proteus* (C) genomes deposited in GenBank and PATRIC were analyzed and drawn into three trees, as shown in panels A, B, and C. Genomes belonging to the same subclade with the closest evolutionary distance are indicated in red-colored text labels. The various phylogenetic clades and subclades are highlighted together and uniquely to show their evolution and epidemiology. The trees were drawn using the maximum-likelihood method in RAxML, using *Streptomyces* sp. MUSC164 as a reference. Bootstrap values are shown on the branches.

The genetic support of the various resistance genes, particularly *bla*$_{CTX-M-15}$, *bla*$_{TEM-1}$, *bla*$_{OXA}$, and *mcr-9.1*, identified in these isolates is not new. In particular, the cooccurrence of *bla*$_{CTX-M-15}$ and *bla*$_{TEM-1}$ within Tn*3* composite transposons, IS*Ec9*, and IS*19* on IncF type plasmids is widely reported in both South Africa and worldwide (4, 20, 21, 29–31). Further, the presence of the *mcr-9.1* gene on IncHI2 plasmids as well as the presence of a cupin fold metalloprotein, as observed here, is common around *mcr-9.1* genes (23). The promiscuity of IncF plasmids and the abundance of integrons, trans-

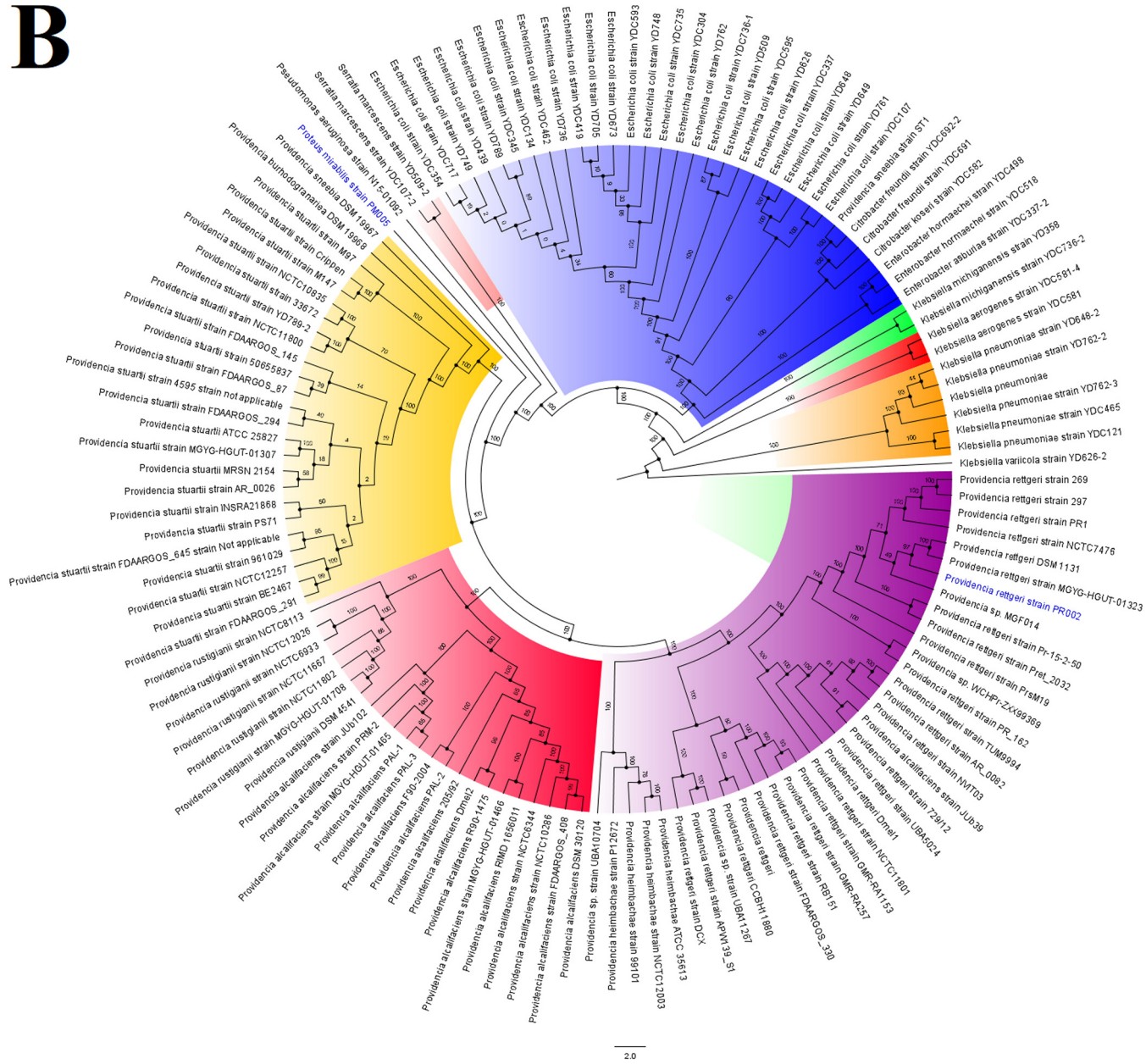

**FIG 6** (Continued)

posons, and insertion sequences (ISs) in the genomes of these strains are worrying, as they might mobilize and facilitate the more rapid dissemination of these multiple resistance genes to other species and clones (4, 31, 32).

It is interesting to note that not all the closely related strains around the *mcr-9.1* strains harbored the *mcr* gene. Some of the global strains with very close phyletic clustering with the non-*mcr*-positive strains contained *mcr* and, in some cases, carbapenemase ($bla_{NDM}$, $bla_{OXA-48}$, $bla_{VIM}$, and $bla_{IMP}$) genes. Such resistome differences between strains belonging to the same clade suggest the gain and loss of resistance plasmids during the evolutionary and epidemiological trajectory. This could be the case for the other *E. hormaechei* strains that had no *mcr* gene. The 11 strains included in this study were phylogenetically distant from any strain from South Africa or Africa but were of very close evolutionary distance to international strains, some of which harbored very rich resistomes, including the cooccurrence of carbapenemase and *mcr* genes.

# C

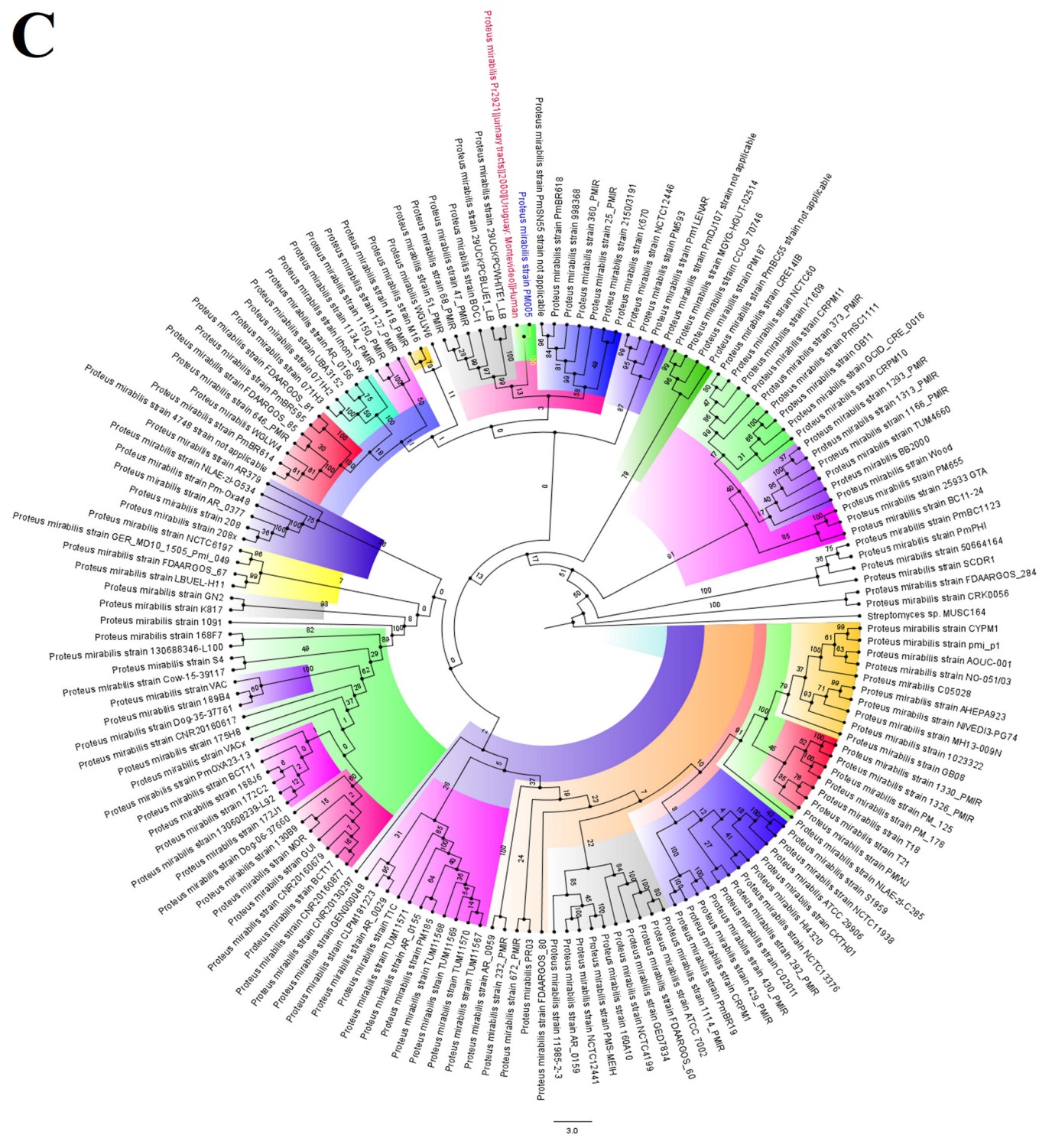

**FIG 6** (Continued)

Thus, the possibility of these strains having been imported cannot be ruled out, a situation that warrants constant surveillance and screening of medical tourists, particularly those from areas of carbapenemase and *mcr* gene endemicity (6, 33).

Although carbapenemases were not found in these strains, as has been reported elsewhere in *mcr*-positive strains (34–38), the cooccurrence of *mcr* and ESBL genes is worrying, as they restrict therapeutic options. To date, only *mcr-1* genes have been

**TABLE 4** Resistome and mobilome characteristics of the isolates

| Plasmid type and sample code (MLST clone) | Species | Resistance genes | Plasmid replicons | pMLST | Integron(s) |
|---|---|---|---|---|---|
| **IncF[F-:A13:B-]** | | | | | |
| EC009 (ST-231) | Enterobacter hormaechei | arr-2::ere(A), aac(3)-IIa, aadA1, strA::strB::sul2, fosA, oqxA::oqxB, sul1::cmlA1, dfrA14, catA2, bla$_{ACT-25}$, catB3::bla$_{OXA-1}$::aac(6')Ib-cr5, bla$_{TEM-1B}$::bla$_{CTX-M-15}$, bla$_{OXA-9}$, aph(3')-Ib, aph(6)-Id, bla$_{LAP-2}$, catA, cmlA5, qacEΔ1, qnrS1 | IncR, ColpVC, ColRNAI | IncF[F-:A13-B-] | In191 |
| EC010 (ST-231) | Enterobacter hormaechei | aph(3')-Ib, aph(6)-Id, aadA1, aac(3)-IIa, qnrB1, fosA, sul1, dfrA14, dfrA1, tet(A), tet(B), bla$_{ACT-41}$::IS2::qnrS1, sul2::strA::bla$_{TEM-1B}$::bla$_{CTX-M-15}$, bla$_{CMY/LAP-2}$, aac(6')Ib-cr5::bla$_{OXA-1}$::catB3, oqxA::oqxB, catA, qacEΔ1, tet(A) | Col(BS512), IncR, Col(MG828), ColpVC, ColRNAI | IncF[F-:A13-B-] | In191 |
| CF003 | Citrobacter freundii | bla$_{CMY-79}$, aac(6')-If | IncA/C2, ColRNAI | IncHI2[ST-1], IncF[F-:A13*:B-] | None |
| **IncHI2[ST-1]** | | | | | |
| CF003 | Citrobacter freundii | bla$_{CMY-79}$, aac(6')-If | IncA/C2, ColRNAI | IncHI2[ST-1], IncF[F-:A13*:B-] | None |
| EC001 (ST-459) | Enterobacter hormaechei | arr-2::ere(A), aadA24, aac(3)-IIa, strA::strB::sul2, sul1::cmlA1, qnrS1, fosA, oqxA::oqxB, dfrA14, catB3, bla$_{OXA-1}$::aac(6')Ib-cr, catA2, bla$_{ACT-7}$, bla$_{OXA-9}$, bla$_{TEM-1B}$::bla$_{CTX-M-15}$ | IncR, ColRNAI | IncHI2[ST-1] | In191, In705, In363 |
| EC015 | Enterobacter hormaechei | aadA1, strB, strA, aac(3)-IIa, sul2, fosA, qnrB1, dfrA14, tet(A), catA1, catB3, bla$_{ACT-16}$, bla$_{CTX-M-15}$, bla$_{OXA-1}$, bla$_{TEM-1B}$, aac(6')Ib-cr | ColRNAI | IncHI2[ST-1] | In191, In705 |
| K006 | Enterobacter hormaechei | aac(3)-II, aac(6')-IIc, aac(6')-Ib3, aadA2, aph(3")-Ib, aph(3')-Ia, aph(6)-Id, arr, bla$_{ACT-56}$, bla$_{SHV-12}$, bla$_{TEM-1}$, catA, catA2, dfrA19, ere(A), fosA, mcr-9.1, qacEΔ1, sul1, sul2, tet(D) | ColRNAI | IncHI2[ST-1], IncF[K1:A-:B-] | In46, In127, In615 |
| K130 (ST-90) | Enterobacter hormaechei | aacA4, aadA2, strA, aph(3')-Ia, aac(6')-IIc, aac(3)-II, aac(6')-Ib3, aph(3")-Ia, aph(6)-Id, arr, bla$_{ACT-56}$, bla$_{ACT-15}$, bla$_{TEM-1}$ catA, catA2, bla$_{TEM-1B}$, bla$_{SHV-12}$, catA, dfrA19, mcr-9.1, qacEΔ1 | ColRNAI | IncHI2[ST-1] | In127, In46, In615 |
| K63 | Enterobacter hormaechei | aacA4, aadA2, aac(3)-IIa, aph(3')-Ia, sul1, sul2, fosA, ere(A), qnrA1, dfrA12, dfrA19, tet(D), catB3, catA2, bla$_{ACT-15}$ bla$_{TEM-1}$ bla$_{CTX-M-15}$ bla$_{OXA-1}$, bla$_{OXA-9}$ aac(6')-IIc, aac(6')-Ib3, aph(3")-Ib, aph(6)-Id, arr, bla$_{ACT-56}$ bla$_{SHV-12}$ mcr-9.1, qacEΔ1 | IncR, ColRNAI | IncF[K2:A13:B-], IncHI2[ST-1] | In127, In46, In615 |
| **IncF[K12:A-:B-]** | | | | | |
| K001 (ST-1791) | Klebsiella variicola | arr-3, fosA, sul2, aacA4, strB, strA, qnrB66, oqxA, frA14, bla$_{CTX-M-15}$, bla$_{LEN13}$, bla$_{TEM-1B}$, aac(6')Ib-cr | IncR, ColRNAI | IncF[K12:A-:B-] | In191, In792 |
| **IncF[K1:A-:B-]** | | | | | |
| K006 | Enterobacter hormaechei | aac(3)-II, aac(6')-IIc, aac(6')-Ib3, aadA2, aph(3")-Ib, aph(3')-Ia, aph(6')-Id, arr, bla$_{ACT-56}$ bla$_{SHV-12}$ bla$_{TEM-1}$, catA, catA2, dfrA19, ere(A), fosA, mcr-9.1, qacEΔ1, sul1, sul2, tet(D) | ColRNAI | IncHI2[ST-1], IncF[K1:A-:B-] | In46, In127, In615 |
| **IncF[K2:A13:B-]** | | | | | |
| K63 | Enterobacter hormaechei | aacA4, aadA2, aac(3)-IIa, aph(3')-Ia, sul1, sul2, fosA, ere(A), qnrA1, dfrA12, dfrA19, tet(D), catB3, catA2, bla$_{ACT-15}$ bla$_{TEM-1}$ bla$_{CTX-M-15}$ bla$_{OXA-1}$, bla$_{OXA-9}$ aac(6')-IIc, aac(6')-Ib3, aph(3")-Ib, aph(6)-Id, arr, bla$_{ACT-56}$ mcr-9.1, qacEΔ1 | IncR, ColRNAI | IncF[K2:A13:B-], IncHI2[ST-1] | In127, In46, In615 |
| **pMLST not detected** | | | | | |
| PM005 | Providencia alcalifaciens | aac(3)-IIa, aadA2, sul2, sul1, dfrA12, tet(J), tet(A), cat, floR, aph(3")-Ib, aph(6)-Id, bla$_{CMY-2}$, bla$_{SCO-1}$ bla$_{TEM-131}$ catA, qacEΔ1 | A/C$_2$ | None | In27 |
| CF004 | Citrobacter freundii | bla$_{CMY-151}$, fosA7 | None | None | None |

reported in South Africa (6, 25, 39, 40), making the identification of *mcr-9.1* in strains as long ago as 2013 a concerning situation. This suggests that other *mcr* variants may be present in South Africa and Africa, and intensive clinical surveillance is necessary to unearth these and preempt further escalation.

## MATERIALS AND METHODS

**Clinical bacterial specimens and antibiotic sensitivity testing.** An old collection of clinical specimens ($n = 72$) that was obtained in 2013 at a referral laboratory from patients having drug-resistant infections at an academic teaching hospital were analyzed (20, 21, 29). These patients suffered from several different diseases, and the specimens were obtained from different body sites. The specimens were cultured on blood agar overnight, and pure colonies were transferred to freshly prepared Mueller-Hinton plates for another 24 h of incubation. The fresh cultures were then used for antimicrobial sensitivity testing and species identification on a MiscroScan Walkaway system. The isolates were further tested for ESBL production using double-disk synergy testing (41). *Enterobacteriaceae* species having reduced sensitivity to cephems and colistin were further selected for genomic analyses.

**Whole-genome sequencing.** Eleven isolates belonging to *Enterobacteriaceae* species that are rarely isolated at the referral laboratory were cultured for 24 h to determine the reservoirs of resistance genes in less-isolated pathogens or commensals. Their genomic DNA was subsequently isolated using the ZR Fungal/Bacterial DNA MiniPrep kit (Zymo Research, USA). Briefly, 200-bp libraries were prepared from the genomic DNA (gDNA) and size-selected using 2% agarose gel and Pippen prep (Sage Science, Beverly, MA, USA). The libraries were barcoded and pooled for sequencing on an Ion Proton system (Thermo Fisher, Waltham, MA, USA).

**Bioinformatic and phylogenomic analyses.** Cutadapt 2.8 was used to remove sequence adapters, artifacts, and poor sequences using default parameters. The SPAdes assembler was used to assemble the trimmed raw reads *de novo* using default parameters. Reads below 200 nucleotides were removed, deposited at GenBank, and annotated with PGAP (42). The resistomes of the isolates were determined from the Isolates Browser database of NCBI (https://www.ncbi.nlm.nih.gov/pathogens/isolates#/search/), while the mobilomes were manually curated from the gff files from GenBank. The isolates' multilocus sequence typing (MLST) sequence types were identified with the MLST database at CGE (http://cge.cbs.dtu.dk/services/MLST/). INTEGRALL (http://integrall.bio.ua.pt/) was used to identify the integrons and gene cassettes within the genomes, while PlasmidFinder (http://cge.cbs.dtu.dk/services/PlasmidFinder/) was used to identify the plasmid replicons.

Contigs bearing the *mcr-9.1* gene were subjected to a BLAST search (using nucleotide BLAST) to identify genomes with the closest nucleotide identity; these genomes were used to draw distance trees using the fast minimum evolution method at a maximum sequence difference of 0.75. The genomes of *Citrobacter freundii*, *Enterobacter hormaechei*, *Klebsiella variicola*, *Providencia* spp., and *Proteus mirabilis* were curated from PATRIC (https://www.patricbrc.org) and filtered to remove poor genomes; i.e., genomes with fewer than 1,000 consensus genes for alignment with the total genome collection were discarded. The filtered genomes were then aligned using CLUSTAL and run through RAxML to draw maximum-likelihood trees (using a bootstrap reassessment of 1,000×), which were annotated using Figtree (http://tree.bio.ed.ac.uk/software/figtree/).

All *mcr-9* genes and selected RefSeq *mcr-1* to *mcr-10* genes were downloaded from GenBank and aligned with MUSCLE. The aligned file was then used to draw a maximum likelihood tree using PhyML. The tree was thereafter annotated using Figtree.

**Data availability.** The genomes of the isolates used in this study have been deposited in DDBJ/ENA/GenBank under BioProject number PRNA355910 and accession numbers NXKE00000000 (CF003), NXKF00000000 (CF004), NXIK00000000 (EC001), NXIL00000000 (EC009), NXIM00000000 (EC010), NXJF00000000 (EC015), NXJG00000000 (K001), NXJH00000000 (K006), NXKO00000000 (K130), NXJN00000000 (K063), and NXKC00000000 (PM005). The versions described in this paper are versions NXKE01000000, NXKF01000000, NXIK01000000, NXIL01000000, NXIM01000000, NXJF01000000, NXJG01000000, NXJH01000000, NXKO01000000, NXJN01000000, and NXKC01000000.

## SUPPLEMENTAL MATERIAL

Supplemental material is available online only.

**FIG S1**, PDF file, 0.03 MB.

**TABLE S1**, XLSX file, 0.02 MB.

**DATA SET S1**, XLSX file, 0.04 MB.

**DATA SET S2**, TXT file, 0.02 MB.

**DATA SET S3**, TXT file, 0.2 MB.

## ACKNOWLEDGMENTS

We declare no conflict of interest.

J.O.S. designed and supervised the study and performed all analyses, bioinformatics, image designs, and writing of the manuscript; N.E.M., L.M., and N.M.M. performed the laboratory assays; all authors proofread and agreed to the final version of the manuscript.

All protocols and consent forms were executed according to the agreed ethical approval terms and conditions. All clinical samples were obtained from a reference laboratory and not directly from patients, who agreed to our using their specimens for this research. The guidelines stated by the Declaration of Helsinki for involving human participants were followed in the study.

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
