## [Reviewer comments · mSystems]

Emergence of *mcr-9.1* in ESBL-producing Clinical Enterobacteriaceae in Pretoria, South Africa: Global Evolutionary Phylogenomics, Resistome and Mobilome.

John Osei Sekyere, Nontuthuko Maningi, Lesedi Modipane, and Nontombi Mbelle

Corresponding Author(s): John Osei Sekyere, University of Pretoria

Review Timeline:

Submission Date:	February 18, 2020
Editorial Decision:	March 6, 2020
Revision Received:	March 7, 2020
Editorial Decision:	March 30, 2020
Revision Received:	March 30, 2020
Editorial Decision:	April 27, 2020
Revision Received:	April 27, 2020
Accepted:	April 28, 2020

Editor: Casey Greene

Reviewer(s): Disclosure of reviewer identity is with reference to reviewer comments included in decision letter(s). The following individuals involved in review of your submission have agreed to reveal their identity: Egon Anderson Ozer (Reviewer #2)

Transaction Report:

DOI: <https://doi.org/10.1128/mSystems.00148-20>

Professor Casey Greene

mSystems

12th February 2020

Dear Professor Greene

RESPONSE TO REVIEWERS' COMMENTS

Please find attached below our response to the reviewers' responses.

Reviewer comments	Responses
Reviewer #1 (Comments for the Author):	
Major comments:	
I am not sure how Figure 2 shows the evolutionary epidemiology of mcr-9. The contig tree that the authors present is, supposedly, based on the whole contig sequence rather than on mcr-9 sequence only? To actually show evolutionary epidemiology, it would probably be best to show a gene tree of mcr 9 vs. a contig tree or a species / strain tree.	A new figure, figure 1B, has been included to show the evolution of only mcr genes, including other mcr variants, to show the evolution of mcr genes, specifically mcr-9. We thank the reviewer for drawing our attention to this. Also see lines 222-229 & 320-331
Minor comments:	
1. P. 9, L. 215: "K006 and K063 were of the same clone". Do they mean clonal isolate? This seems to be in contrast with P. 5, and P. 8 (line 190) I can only assume a typo here, otherwise, I'm confused.	No, it is not in contrast, neither is it a typographical error. Let me explain. when isolates share very close sequence similarity, they can be classified into the same clade or clone. Such strains originate from the same ancestor and are spread from patient to patient via a healthcare worker, direct contact or the hospital environment. This could have been the case here where strains of the same clone were found in two different patients from the same hospital. See lines 315-318
P. 11, L. 272-273: "the contigs bearing the mcr-9.1 genes were not of the same homology". Not of the same sequence homology to what?	When aligned to themselves, the 3 mcr-9 contigs did not have 100% sequence homology to each other. See line 309
P 12 L 286 "Salmonella Typhimurium"  "Salmonella typhimurium" (all italics).	Salmonella enterica subsp. enterica ser. Typhimurium is shortened as Salmonella Typhimurium or S. Typhimurium . Please note that the 'Typhimurium' is not a species

	name, but the serovar. Hence, it is not italicized. This is the conventional scientific nomenclature.
Reviewer #2 (Comments for the Author):	
Page 4, line 87: Why were rarely-isolates species of Enterobacteriaceae selected for this study? It's unclear what the clinical significance of identifying antimicrobial resistance mechanisms in these species is relative to other more common and potentially more pathogenic species.	Thank you for drawing our attention to this. We have provided reasons in lines 69-76 and 99-100.
Page 5, lines 103 - 110. The bioinformatic methods are vague and difficult to judge without more detail (i.e. filter parameters, etc.)	Please note that most of the bioinformatic tools used are web-based free software that come with default parameters. A filtering/trimming step with Cutadapt has been added. Further removal of reads with less than 200nt were also removed. Assembly with SPAdes was with default parameters. Genome annotations were done by NCBI and the web-based applications used for the other analyses have been clearly stated. The phylogenetic analyses steps have also been clearly defined. Please see lines 106-128. If there are other missing parameters, we humbly call on the reviewer to point it out.
Figure 1 is unnecessary. Table 1 is a better presentation of the data.	Thank you. Figure 1 has been removed.
Table 2 is confusing. It is difficult to understand which mutations were found in which isolates.	Thank you. Table 2 has been revised.
Page 6, line 149: It is not explained how the genetic environment suggests that the AmpC genes are found on the chromosomes.	The contigs on which the AmpCs were based were BLASTed on GenBank and they aligned most closely with chromosomes on GenBank. This is shown in lines 168-170
All of the tree figures are illegible without significant zooming in. The colors of the taxon labels are difficult to see.	We have tried to increase the resolution of all the figures. We also placed all the figures into individual canvases to increase the visibility. Further increasing the sizes of the images will make it blurred and too big; thus, this is the best we can do so far.

Page 8, line 185: There does not appear to be a Figure 2C.	The figures have been revised now. Figure 2C is now included
Page 12, line 286: It is shown in this report that only one of the three mcr9.1-containing genomes was colistin resistant. Although I agree that antibiotic use should be carefully applied and limited, in the context of the incomplete correlation of mcr-9.1 with resistance in these findings, how does this conclusion follow your results? More explanation may be useful.	The absence of resistance in mcr strains is not new and does not suggest that their detection is not important as they can be highly expressed to cause resistance under certain promoters. See lines 300-306
Page 13, line 316: The strains reported here were isolated 7 years ago from 2 institutions. It is unclear, as stated here, that they represent "emergence" of mcr-9.1 in South Africa. More context for this statement would help.	The word emergence has been changed to "identification" in line 369. Please note that although mcr-9.1 was identified in isolates that are as old as 2013, the detection of the gene in 2020 represents the first report of this gene in Africa, making it an 'emerging' problem.

March 6, 2020

Dr. John Osei Sekyere
University of Pretoria
Department of Medical Microbiology
School of Medicine
Faculty of Health Sciences
Pretoria, Gauteng 0084
South Africa

Re: mSystems00148-20 (Emergence of mcr-9.1 in ESBL-producing Clinical Enterobacteriaceae in Pretoria, South Africa: Global Evolutionary Phylogenomics, Resistome and Mobilome.)

Dear Dr. John Osei Sekyere:

The reviewers found your manuscript substantially improved, but have some remaining points that will improve the clarity of your submission.

Please note that ASM has a Data Policy (<https://journals.asm.org/content/open-data-policy>). It appears that, according to the policy, the whole genome sequencing data should be submitted to a repository such as EBI's ENA.

Below you will find the comments of the reviewers.

To submit your modified manuscript, log onto the eJP submission site at <https://msystems.msubmit.net/cgi-bin/main.plex>. If you cannot remember your password, click the "Can't remember your password?" link and follow the instructions on the screen. Go to Author Tasks and click the appropriate manuscript title to begin the resubmission process. The information that you entered when you first submitted the paper will be displayed. Please update the information as necessary. Provide (1) point-by-point responses to the issues raised by the reviewers as file type "Response to Reviewers," not in your cover letter, and (2) a PDF file that indicates the changes from the original submission (by highlighting or underlining the changes) as file type "Marked Up Manuscript - For Review Only."

Please return the manuscript within 60 days; if you cannot complete the modification within this time period, please contact me. If you do not wish to modify the manuscript and prefer to submit it to another journal, please notify me of your decision immediately so that the manuscript may be formally withdrawn from consideration by mSystems.

To avoid unnecessary delay in publication should your modified manuscript be accepted, it is important that all elements you upload meet the technical requirements for production. I strongly recommend that you check your digital images using the Rapid Inspector tool at <http://rapidinspector.cadmus.com/RapidInspector/zmw/>.

If your manuscript is accepted for publication, you will be contacted separately about payment when the proofs are issued; please follow the instructions in that e-mail. Arrangements for payment must be made before your article is published. For a complete list of **Publication Fees**, including

supplemental material costs, please visit our website.

Sincerely,

Casey Greene

Editor, mSystems

Journals Department
Reviewer comments:

Reviewer #1 (Comments for the Author):

I thank the authors for addressing the previous comments. A few issues still remain:

1. The authors seem to confuse "homology" with "sequence identity", which leads to somewhat confusing statements in the manuscript. Homology is not a quantitative trait, but a binary one. Two sequences are homologous if derived from a common ancestor, and not homologous if they are not. Therefore, statements such as "100% homology" are incorrect. At the same time, percent sequence identity is an indicator of homology, when the percent identity is high enough. For more, see:

"Homology in Proteins and Nucleic Acids: A Terminology Muddle and a Way out of It" (Cell 1987)
[https://www.cell.com/cell/pdf/0092-8674\(87\)90322-9.pdf](https://www.cell.com/cell/pdf/0092-8674(87)90322-9.pdf) and

"An Introduction to Sequence Similarity ("Homology") Searching" Current Protocols 2013

<https://currentprotocols.onlinelibrary.wiley.com/doi/full/10.1002/0471250953.bi0301s42>

In light of this, the following corrections are needed:

171: replace "homology" with "percent identity", also "aligned closely with only chromosomes" does not exactly make sense. What does "aligned closely" mean?

311: "100% nucleotide sequence homology" replace with "identity"

327: again "percent sequence identity" instead of "homology"

329: cannot have a "closer" homolog, since homology is binary. A "more recent common ancestor" would probably be the term that should be used.

"close sequence similarity" best replaced with "high sequence similarity" If these are nucleic acid sequences, "sequence identity" is probably better used over "similarity".

2. In the phylogenetic trees, it is hard to see where the bootstrap values are significant, and where not. The authors claim that (line 125) they bootstrapped the trees, but the values are not apparent in the figures. Some indication of significance in the proper inner nodes (such as an asterisk) would be needed.

Reviewer #2 (Comments for the Author):

Thank you for the thoughtful responses to my comments. The changes made satisfy my concerns with only the following note:

On line 72-73 in the revised manuscript, you list "*Klebsiella pneumoniae*" as a "rarely-isolated pathogenic species." Did you mean to write "*Klebsiella variicola*" or "*oxytoca*" or another less common *Klebsiella* species than *pneumoniae*?

Professor Casey Greene

mSystems

6th March 2020

Dear Professor Greene

RESPONSE TO REVIEWERS' COMMENTS

Please find attached below our response to the reviewers' responses.

Reviewer comments	Responses
Reviewer #1 (Comments for the Author):	
1. I thank the authors for addressing the previous comments. A few issues still remain:	Thank you very much. We also appreciate your constructive comments
2. The authors seem to confuse "homology" with "sequence identity", which leads to somewhat confusing statements in the manuscript. Homology is not a quantitative trait, but a binary one. Two sequences are homologous if derived from a common ancestor, and not homologous if they are not. Therefore, statements such as "100% homology" are incorrect. At the same time, percent sequence identity is an indicator of homology, when the percent identity is high enough. For more, see: "Homology in Proteins and Nucleic Acids: A Terminology Muddle and a Way out of It" (Cell 1987) https://www.cell.com/cell/pdf/0092-8674(87)90322-9.pdf and "An Introduction to Sequence Similarity ("Homology") Searching" Current Protocols 2013 https://currentprotocols.onlinelibrary.wiley.com/doi/full/10.1002/0471250953.bi0301s42	Thank you very much. We highly appreciate your educative comments. We shall study further into it and amend our terms going forward.
3. In light of this, the following corrections are needed:	
171: replace "homology" with "percent identity", also "aligned closely with only chromosomes" does not exactly make sense. What does "aligned closely" mean?	This has been done and highlighted throughout the text.
311: "100% nucleotide sequence homology" replace with "identity"	
327: again "percent sequence identity" instead of "homology"	
329: cannot have a "closer" homolog, since homology is binary. A "more recent common ancestor" would probably be the term that should be used.	
"close sequence similarity" best replaced with "high sequence similarity" If these are	

nucleic acid sequences, "sequence identity" is probably better used over "similarity".	
2. In the phylogenetic trees, it is hard to see where the bootstrap values are significant, and where not. The authors claim that (line 125) they bootstrapped the trees, but the values are not apparent in the figures. Some indication of significance in the proper inner nodes (such as an asterisk) would be needed.	I did not include the bootstraps because they were obstructing the branches and marring the beauty of the trees, making it difficult to see the nodes & branches. I have however annotated the branches around the strains/ genes of interest with bootstraps above 0.5
Reviewer #2 (Comments for the Author):	
Thank you for the thoughtful responses to my comments. The changes made satisfy my concerns with only the following note:	Thank you very much. We also appreciate your constructive comments
On line 72-73 in the revised manuscript, you list "Klebsiella pneumoniae" as a "rarely-isolated pathogenic species." Did you mean to write "Klebsiella variicola" or "oxytoca" or another less common Klebsiella species than pneumoniae?	Thanks for drawing our attention to this. I

	changed this to K. variicola
--	--

March 30, 2020

Dr. John Osei Sekyere
University of Pretoria
Department of Medical Microbiology
School of Medicine
Faculty of Health Sciences
Pretoria, Gauteng 0084
South Africa

Re: mSystems00148-20R1 (Emergence of mcr-9.1 in ESBL-producing Clinical Enterobacteriaceae in Pretoria, South Africa: Global Evolutionary Phylogenomics, Resistome and Mobilome.)

Dear Dr. John Osei Sekyere:

Please see the comments from the first reviewer. The advice to consult with a scientist with expertise in phylogenomics seems well-founded to report appropriate statistical confidences related to the trees.

Below you will find the comments of the reviewers.

To submit your modified manuscript, log onto the eJP submission site at <https://msystems.msubmit.net/cgi-bin/main.plex>. If you cannot remember your password, click the "Can't remember your password?" link and follow the instructions on the screen. Go to Author Tasks and click the appropriate manuscript title to begin the resubmission process. The information that you entered when you first submitted the paper will be displayed. Please update the information as necessary. Provide (1) point-by-point responses to the issues raised by the reviewers as file type "Response to Reviewers," not in your cover letter, and (2) a PDF file that indicates the changes from the original submission (by highlighting or underlining the changes) as file type "Marked Up Manuscript - For Review Only."

Due to the SARS-CoV-2 pandemic, our typical 60 day deadline for revisions will not be applied. I hope that you will be able to submit a revised manuscript soon, but want to reassure you that the journal will be flexible in terms of timing, particularly if experimental revisions are needed. When you are ready to resubmit, please know that our staff and Editors are working remotely and handling submissions without delay. If you do not wish to modify the manuscript and prefer to submit it to another journal, please notify me of your decision immediately so that the manuscript may be formally withdrawn from consideration by mSystems.

To avoid unnecessary delay in publication should your modified manuscript be accepted, it is important that all elements you upload meet the technical requirements for production. I strongly recommend that you check your digital images using the Rapid Inspector tool at <http://rapidinspector.cadmus.com/RapidInspector/zmw/>.

If your manuscript is accepted for publication, you will be contacted separately about payment when the proofs are issued; please follow the instructions in that e-mail. Arrangements for payment must be made before your article is published. For a complete list of **Publication Fees**, including

supplemental material costs, please visit our website.

Sincerely,

Casey Greene

Editor, mSystems

Journals Department
Reviewer comments:

Reviewer #1 (Comments for the Author):

"Nodes with bootstrap figures above 50% (significant) are annotated with a filled black circle."

I am curious to know why the claim that 50% is a significant bootstrap value, when, by definition, it means that there is a 50% chance of a Type I error, no better than a coin toss.

100x bootstrap is not a sufficient sample size in any case, given the large number of leaf nodes in the trees. Based on their presentation, I see no evidence that the sequence data supports the tree topologies.

I would suggest the authors consult with a phylogeneticist, or with a basic bootstrapping guide text, to provide a meaningful tree that includes proper sampling size for the bootstrap, and significant values.

Significant bootstrap values can be marked with dots of varying shapes and sizes, so as to maintain tree aesthetics.

Professor Casey Greene

mSystems

30th March 2020

Dear Professor Greene

RESPONSE TO REVIEWERS' COMMENTS

Please find attached below our response to the reviewers' responses.

Reviewer comments	Responses
Reviewer #1 (Comments for the Author):	
"Nodes with bootstrap figures above 50% (significant) are annotated with a filled black circle."	Explained below.
I am curious to know why the claim that 50% is a significant bootstrap value, when, by definition, it means that there is a 50% chance of a Type I error, no better than a coin toss.	Agreed. This has been removed.
100x bootstrap is not a sufficient sample size in any case, given the large number of leaf nodes in the trees. Based on their presentation, I see no evidence that the sequence data supports the tree topologies.	Sorry. This was a typographical error. The bootstrap sampling was done 1000x according to the default parameters of the application used.
I would suggest the authors consult with a phylogenticicist, or with a basic bootstrapping guide text, to provide a meaningful tree that includes proper sampling size for the bootstrap, and significant values.	The earlier bootstrap value was provided in error. The trees have been redrawn with appropriate bootstrap values showing.
Significant bootstrap values can be marked with dots of varying shapes and sizes, so as to maintain tree aesthetics.	Actual bootstrap values have now been provided for all the trees except for Figures 1B & 2A-C, for which Figtree could not show its bootstrap values. For Fig. 1B, it's bootstraps are shown in Fig. S1 as Figtree could not show its bootstraps. Bootstrap analyses were not applied in Figures 2: "Contigs bearing the mcr9.1 genes were BLASTed (using nucleotide BLAST) to identify genomes with closest nucleotide identity, which were used to draw distance trees

	using the Fast Minimum Evolution method at a Maximum Sequence Difference of 0.75”.
--	--

April 27, 2020

Dr. John Osei Sekyere
University of Pretoria
Department of Medical Microbiology
School of Medicine
Faculty of Health Sciences
Pretoria, Gauteng 0084
South Africa

Re: mSystems00148-20R2 (Emergence of mcr-9.1 in ESBL-producing Clinical Enterobacteriaceae in Pretoria, South Africa: Global Evolutionary Phylogenomics, Resistome and Mobilome.)

Dear Dr. John Osei Sekyere:

As you report sequencing 200bp libraries on an Ion Proton, we expect the associated nucleic acid sequence reads will be deposited at ENA, SRA, or a similar widely-used repository for sequencing data and made available for inspection by interested readers. As noted on ASM Journals' Data Policy page under "mSystems open data policy," a paragraph dedicated to new accession numbers for nucleotide and amino acid sequences, microarray data, protein structures, gene expression data, and MycoBank data should appear at the end of Materials and Methods with the paragraph lead-in "Data availability." Please also provide references (with URLs) for the accession numbers.

I am willing to accept the paper assuming that the data is made available and a data availability section is described. Once this is available and submitted I can accept the paper.

Below you will find the comments of the reviewers.

To submit your modified manuscript, log onto the eJP submission site at <https://msystems.msubmit.net/cgi-bin/main.plex>. If you cannot remember your password, click the "Can't remember your password?" link and follow the instructions on the screen. Go to Author Tasks and click the appropriate manuscript title to begin the resubmission process. The information that you entered when you first submitted the paper will be displayed. Please update the information as necessary. Provide (1) point-by-point responses to the issues raised by the reviewers as file type "Response to Reviewers," not in your cover letter, and (2) a PDF file that indicates the changes from the original submission (by highlighting or underlining the changes) as file type "Marked Up Manuscript - For Review Only."

Due to the SARS-CoV-2 pandemic, our typical 60 day deadline for revisions will not be applied. I hope that you will be able to submit a revised manuscript soon, but want to reassure you that the journal will be flexible in terms of timing, particularly if experimental revisions are needed. When you are ready to resubmit, please know that our staff and Editors are working remotely and handling submissions without delay. If you do not wish to modify the manuscript and prefer to submit it to another journal, please notify me of your decision immediately so that the manuscript may be formally withdrawn from consideration by mSystems.

To avoid unnecessary delay in publication should your modified manuscript be accepted, it is

important that all elements you upload meet the technical requirements for production. I strongly recommend that you check your digital images using the Rapid Inspector tool at <http://rapidinspector.cadmus.com/RapidInspector/zmw/>.

Sincerely,

Casey Greene

Editor, mSystems

Journals Department
Reviewer comments:

Reviewer #1 (Comments for the Author):

no comments.

Professor Casey Greene

mSystems

27th April 2020

Dear Professor Greene

RESPONSE TO EDITOR'S COMMENTS

“As you report sequencing 200bp libraries on an Ion Proton, we expect the associated nucleic acid sequence reads will be deposited at ENA, SRA, or a similar widely-used repository for sequencing data and made available for inspection by interested readers. As noted on ASM Journals' Data Policy page under "mSystems open data policy," a paragraph dedicated to new accession numbers for nucleotide and amino acid sequences, microarray data, protein structures, gene expression data, and MycoBank data should appear at the end of Materials and Methods with the paragraph lead-in "Data availability." Please also provide references (with URLs) for the accession numbers.”

Response: These have been provided in lines 132-149.

Thank you.

Sincere regards,

John Osei Sekyere, Ph.D.

April 28, 2020

Dr. John Osei Sekyere
University of Pretoria
Department of Medical Microbiology
School of Medicine
Faculty of Health Sciences
Pretoria, Gauteng 0084
South Africa

Re: mSystems00148-20R3 (Emergence of mcr-9.1 in ESBL-producing Clinical Enterobacteriaceae in Pretoria, South Africa: Global Evolutionary Phylogenomics, Resistome and Mobilome.)

Dear Dr. John Osei Sekyere:

Your manuscript has been accepted, and I am forwarding it to the ASM Journals Department for publication. For your reference, ASM Journals' address is given below. Before it can be scheduled for publication, your manuscript will be checked by the mSystems senior production editor, Ellie Ghatineh, to make sure that all elements meet the technical requirements for publication. She will contact you if anything needs to be revised before copyediting and production can begin. Otherwise, you will be notified when your proofs are ready to be viewed.

Sincerely,

Casey Greene
Editor, mSystems

Journals Department
Supplemental dataset 2: Accept

Supplemental Table S1: Accept

Supplemental Material/dataset 3: Accept

Fig. S1: Accept

Supplemental dataset 1: Accept